# Adaptive and multifunctional hydrogel hybrid probes for long-term sensing and modulation of neural activity

Seongjun Park [1,2,12,13,14], Hyunwoo Yuk [3,14], Ruike Zhao[3,4], Yeong Shin Yim[5,6], Eyob W. Woldeghebriel[1], Jeewoo Kang[7], Andres Canales[2,8], Yoel Fink [2,8,9], Gloria B. Choi [6,10], Xuanhe Zhao [3,11✉] & Polina Anikeeva [2,5,8✉]

To understand the underlying mechanisms of progressive neurophysiological phenomena, neural interfaces should interact bi-directionally with brain circuits over extended periods of time. However, such interfaces remain limited by the foreign body response that stems from the chemo-mechanical mismatch between the probes and the neural tissues. To address this challenge, we developed a multifunctional sensing and actuation platform consisting of multimaterial fibers intimately integrated within a soft hydrogel matrix mimicking the brain tissue. These hybrid devices possess adaptive bending stiffness determined by the hydration states of the hydrogel matrix. This enables their direct insertion into the deep brain regions, while minimizing tissue damage associated with the brain micromotion after implantation. The hydrogel hybrid devices permit electrophysiological, optogenetic, and behavioral studies of neural circuits with minimal foreign body responses and tracking of stable isolated single neuron potentials in freely moving mice over 6 months following implantation.

[1] Department of Electrical Engineering and Computer Science, Massachusetts Institute of Technology, Cambridge, MA, USA. [2] Research Laboratory of Electronics, Massachusetts Institute of Technology, Cambridge, MA, USA. [3] Department of Mechanical Engineering, Massachusetts Institute of Technology, Cambridge, MA, USA. [4] Department of Mechanical and Aerospace Engineering, The Ohio State University, Columbus, OH, USA. [5] McGovern Institute for Brain Research, Massachusetts Institute of Technology, Cambridge, MA, USA. [6] Department of Brain and Cognitive Sciences, Massachusetts Institute of Technology, Cambridge, MA, USA. [7] Department of Chemical Engineering, Massachusetts Institute of Technology, Cambridge, MA, USA. [8] Department of Material Science and Engineering, Massachusetts Institute of Technology, Cambridge, MA, USA. [9] Institute for Soldier Nanotechnologies, Massachusetts Institute of Technology, Cambridge, MA, USA. [10] Picower Institute for Learning and Memory, Massachusetts Institute of Technology, Cambridge, MA, USA. [11] Department of Civil and Environmental Engineering, Massachusetts Institute of Technology, Cambridge, MA, USA. [12] Present address: Department of Bio and Brain Engineering, Korea Advanced Institute of Science and Technology (KAIST), Daejeon, Republic of Korea. [13] Present address: KAIST Institute for Health and Science Technology (KIHST), Daejeon, Republic of Korea. [14] These authors contributed equally: Seongjun Park, Hyunwoo Yuk. ✉email: zhaox@mit.edu; anikeeva@mit.edu

Extending electrophysiological and optogenetics studies of brain circuits from days to years may illuminate neural dynamics underlying progressive neurological disorders[1,2]. However, prolonged implantation of devices into the brain is frequently accompanied by the foreign body response[3,4] hallmarked by the gradual encapsulation of the implanted probes by glial scars and subsequent loss of recording and stimulation capabilities[5]. These adverse reactions have been hypothesized to stem from chronic tissue damage, in part, due to the mismatch in mechanical and chemical properties between the implanted probes and the neural tissue[6]. Despite substantial advances in the development of compliant neural probes targeting the brain[7–9], spinal cord[10], and peripheral nerves[11], the majority of these devices rely on mechanically dissimilar materials with bending stiffness and elastic moduli significantly exceeding those of the neural tissue. Although softer materials such as thermoplastics (e.g., polycarbonate)[12] or elastomers (e.g., polydimethylsiloxane)[11,13] have been recently leveraged to produce compliant neural probes, the elastic moduli of these materials (e.g., 1 MPa to 10 GPa) still surpass those of neural tissue (e.g., 1–10 kPa)[14] by 3–6 orders of magnitude.

Owing to their chemo-mechanical similarity to biological tissues, hydrogels have shown promise as interfaces between biological and synthetic systems[15]. Although hydrogels have already been explored for electrical[16], fluidic[17], and optical[18] bio-interfaces, the intrinsic limitations of these materials such as poor electrical conductivity[16] (typically <1 S cm$^{-1}$ in physiological conditions), non-selective signal dissipation through electrolytic hydrogel medium, and low optical transmission[19] (refractive index 1.33–1.51 that is similar or lower than that of the neural tissue) hinder their sole use in the development of miniaturized multifunctional neural probes. As an alternative strategy, hydrogels have been introduced as exterior coatings for neural probes composed of stiff materials[20]. While these early studies have shown a promising enhancement in biocompatibility following probe implantation[21], several challenges remain to be addressed. Bulk exterior hydrogel coatings on existing probes fail to optimize their device-level mechanical properties and interactions with neural tissues. First, the overall probe dimensions can be significantly increased by the introduction of a bulk hydrogel layer. Second, the combined mechanical properties (e.g., bending stiffness) of these hydrogel-coated devices remain dominated by the stiff materials constituting the original probes. These factors can adversely contribute to acute and chronic tissue–probe interactions, limiting possible benefits from the use of hydrogels. We reasoned that these challenges could, in part, be met by rational biomechanically-inspired design and intimate integration of the hydrogels within neural probe architectures.

Here, we present a hybrid multifunctional probe for long-term neural sensing and actuation of neural activity produced by intimate integration of microscale polymer-based fibers within a soft hydrogel matrix. By combining the fiber drawing process and a robust hydrogel bonding technique, we develop a compliant all-in-one device capable of long-term recording of neural activity, delivery of light, and infusion of chemicals into the mouse brain. The mechanical analysis is applied to design a probe with the low bending stiffness, that significantly reduces the stress field in the surrounding brain tissues during micromotion offering a "stealthy" interface. Furthermore, the adaptive bending stiffness of these probes, rooted in the three orders of magnitude difference in elastic moduli between the dried and fully hydrated hydrogels, enables their direct insertion into deep brain structures without guiding fixtures, while enhancing the probe long-term biocompatibility. We evaluate the performance of these adaptive hydrogel hybrid probes in opto-electrophysiological and behavioral experiments in freely moving mice. Consistent with minimal foreign-body reaction, as revealed by immunohistology, these stealthy probes enable tracking of isolated single-neuron action potentials over 6 months (168 days) following implantation. We envision that this technology will offer opportunities in neuroscience and engineering fields including opto-electrophysiological mapping of circuits, involved in progressive neurological disorders and the development of future bioelectronic medicines.

## Results

**Design, fabrication, and characterization of multifunctional hydrogel hybrid probes**. To take advantage of the hydrogels' mechanical and chemical properties without compromising functional probe performance, we integrated an assembly of microscale polymer fibers within a hydrogel matrix. Unlike bulk hydrogel coatings on the surfaces of fully assembled neural probes[20], we employed hydrogel as a matrix to hybridize multiple individual functional fibers into a multifunctional probe. This composite structure leverages hydrogel's favorable mechanical and chemical properties to reduce the impact on the surrounding tissue, while offering a range of capabilities desirable for systems neuroscience studies (Fig. 1a). To ensure long-term reliability of the hydrogel hybrid probes we apply recent insights from mechanically robust hydrogels and techniques for their bonding to engineering materials[22].

To produce hydrogel hybrid probes, we first fabricate individual functional fibers via thermal drawing (Fig. 1b, d). Microscale optical fibers consisting of polycarbonate (PC) core and cyclic olefin copolymer (COC) cladding, microelectrode arrays containing seven tin (Sn) microwires encapsulated within poly(etherimide) (PEI) insulating cladding, and PEI tubes for microfluidic channels are produced from macroscale preforms at a yield of hundreds of meters (Supplementary Fig. 1), reducing the cost and labor barrier for the production of microscale devices tailored to a specific application.

We subsequently integrate the functional polymer fibers into the soft hydrogel matrix (Fig. 1c and Supplementary Fig. 2). A polyimide guide fixture (Supplementary Fig. 3a) is used to align the fibers into an assembly containing one waveguide (105.9 ± 8.0 μm diameter) at the center, three microelectrode arrays (80.0 ± 1.8 μm diameter, seven 4.75 ± 2.22 μm electrodes), and three microfluidic channels (54.0 ± 2.1 μm inner and 115.4 ± 3.0 μm outer diameters) in an alternating concentric arrangement. Following initial alignment, functional fibers are connected to optical ferrules, electrical pin connectors, and fluidic tubing, which is then followed by the hydrogel integration (Supplementary Fig. 4). To achieve favorable mechanical and chemical interactions with the surrounding brain tissue while allowing robust integration of the assembled functional fibers, a hydrogel matrix for the hybrid probe should meet design criteria including softness, biocompatibility, stability in physiological environments, and strong adhesion to the functional polymer fibers. We characterize mechanical properties (shear moduli and fracture toughness) and interfacial integration with polymer substrates (interfacial toughness on PC) of various biocompatible hydrogel candidates including poly(vinyl alcohol) (PVA), alginate, poly(ethylene glycol) diacrylate-alginate (PEGDA-Alg), and poly(acrylamide)-alginate (PAAm-Alg) (Supplementary Figs. 5 and 6). While all abovementioned hydrogels exhibit low shear moduli below 10 kPa (Supplementary Fig. 5a), PVA, alginate, and PEGDA-Alg hydrogels provide low fracture toughness (<150 J m$^{-2}$) and low interfacial toughness on PC (<20 J m$^{-2}$) compared to PAAm-Alg hydrogel (Supplementary Figs. 5b and 6). Furthermore, the weak interfacial integration between PC and PVA, alginate, and PEGDA-Alg hydrogels results in delamination of the polymer-based functional fibers from the hydrogel matrix within the hybrid probes (Supplementary Table 1).

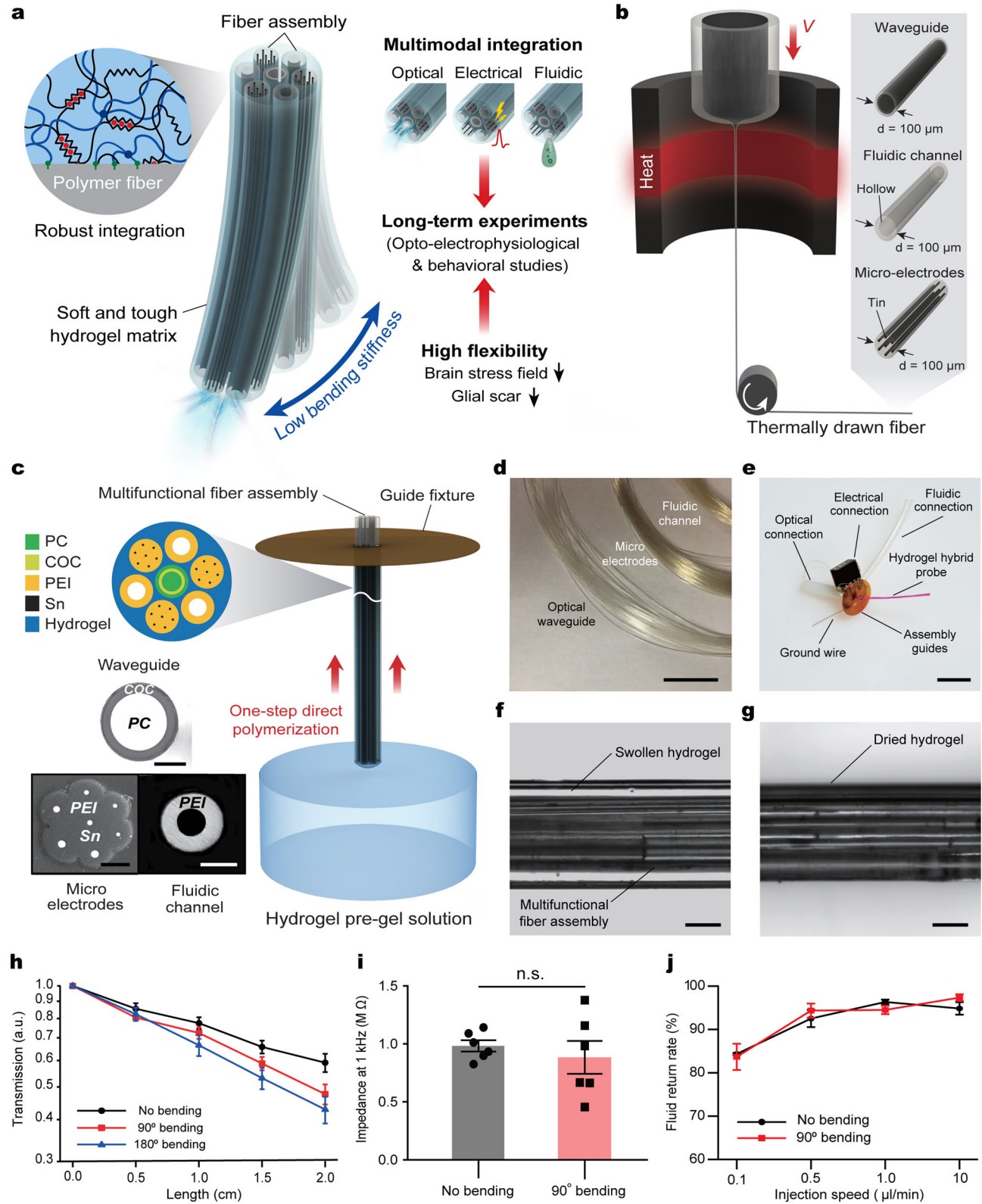

Therefore, PAAm-Alg hydrogel (Supplementary Fig. 7a) is selected as the matrix for the hybrid probes due to its brain tissue-like softness (i.e., 5.5 kPa shear modulus), mechanical robustness (i.e., 500 J m$^{-2}$ fracture toughness)[23], biocompatibility[24], long-term chemical stability in physiological environments (Supplementary Fig. 8), and robust interfacial integration with polymers constituting functional fibers (Supplementary Fig. 9). The latter is achieved by applying a tough bonding technique[22]. Surfaces of the fibers aligned within a guide fixture are functionalized with primary amine groups[25], which then permits covalent grafting of alginate via carbodiimide chemistry to generate robust bonding between the hydrogel matrix and the polymer fibers (Supplementary Fig. 9)[22,26]. After functionalization, the pre-gel solution is dip-coated around the polymer fibers and then subjected to

**Fig. 1 Design, fabrication, and characterization of multifunctional hydrogel hybrid probes. a** A conceptual illustration of the hydrogel hybrid probe design and its application to minimize impact on brain tissue. **b, c** Fabrication of the hydrogel hybrid probe including thermal drawing of the functional fiber units (**b**), and one-step direct polymerization of the hydrogel matrix within the fiber assembly (**c**). Scale bars: 50 μm. **d** A photograph of the optical waveguide, micro-electrode array, and microfluidic channel fibers. Scale bar: 5 cm. **e** A photograph of a hydrogel hybrid probe after integration of the hydrogel matrix with a multifunctional fiber assembly. Hydrogel is dyed with Rhodamine B for visual clarity. Scale bar: 1 cm. **f, g** Microscope images of the hydrogel hybrid probe with a fully swollen (**f**) and a dehydrated (**g**) hydrogel matrix. Scale bars: 100 μm. **h** Optical transmission losses of the PC/COC waveguides within the hydrogel hybrid probes at 0°, 90°, and 180° bending deformation. **i** Tip impedance of the electrodes within the fiber arrays in the hydrogel hybrid probes at 0° and 90° bending (paired two-sided Student's $t$-test: $p = 0.5232$). **j** Return rate of the microfluidic channel fibers within the hydrogel hybrid probes at 0° and 90° bending deformation. Values in **h**–**j** represent the mean and the standard deviation ($n = 6$).

one-step polymerization under ultraviolet irradiation (254 -> 365 nm wavelength, 8 W) (Fig. 1e). To avoid bulk hydrogel layer formation around the hybrid probes, any excessive pre-gel solution on the probe is scrapped off by a polyimide scraper prior to polymerization (Supplementary Fig. 3b) to ensure the uniform thickness (25 μm) of hydrogel matrix around the polymer fiber assembly. The hydrogel thickness (25 μm) in the hybrid probes is selected to minimize the size of the devices, while providing sufficient mechanical cloaking for the underlying stiffer polymer fibers[27].

The bonding between the functionalized fibers and the PAAm-Alg hydrogel matrix exhibits interfacial toughness of 230 J m$^{-2}$, which is significantly greater than the interfacial toughness without functionalization (5 J m$^{-2}$) (Supplementary Fig. 10). The resulting probes exhibit intimate integration between the hydrogel matrix and the fiber assembly, with a uniform and thin outermost hydrogel layer across the probes and low probe-to-probe variability (Fig. 1f and Supplementary Fig. 11b). The probes are further soaked in phosphate buffered saline for 3 days to eliminate any unreacted reagents. High performance liquid chromatography (HPLC) analysis of PAAm-Alg hydrogels shows that this thorough cleaning process can effectively remove residual monomers from the hydrogel matrix (Supplementary Fig. 7b, c). Notably, the integrity of the hydrogel matrix and its bonding to the polymer fibers within the hybrid probe is still maintained even in a fully dehydrated state (Fig. 1g). The optical (Fig. 1h), electrical (Fig. 1i), and fluidic (Fig. 1j) properties of the hybrid devices are not significantly affected by 90° and 180° bending, and the repeated deformation cycles (10,000 times) do not induce significant changes in these properties (Supplementary Fig. 12).

**Mechanical analysis of hydrogel hybrid probes and probe–tissue interactions.** While optical, electrical, and fluidic characteristics determine the probe functional performance, the mechanical properties are anticipated to dominate the probe–tissue interactions during implantation and long-term use. Hence, we quantitatively evaluate the mechanical properties of the hydrogel hybrid probes using finite element analysis (FEA) (Supplementary Movie 1, 2). Due to the cylindrical geometry of the hybrid probes and low adhesion between biological tissues and PAAm-Alg hydrogels[24], mechanical interactions between the hybrid probes and the surrounding brain tissue are predominantly determined by the bending of the probes (with negligible contributions from torsion). Hence, bending stiffness of the hydrogel hybrid probes is the key mechanical parameter defining the probe–tissue interactions in our study. The hybrid probe with a fully swollen hydrogel matrix exhibits a bending stiffness (7 N m$^{-1}$) significantly lower than that of fibers composed of stainless steel (8191 N m$^{-1}$), silica (2150 N m$^{-1}$), and PC (103 N m$^{-1}$) with identical dimensions (334 μm diameter and 3.4 mm length) (Fig. 2a, Supplementary Table 2, and Supplementary Movie 1). Notably, the hydrogel hybrid probe exhibits ~15 times lower bending stiffness than the PC probe, despite

~65% of the hybrid probe's volume being occupied by the same polymer. This difference in bending stiffness stems from the bundled-fiber design of the hydrogel hybrid probes. During bending deformation, the soft hydrogel matrix with Young's modulus six orders of magnitude lower than that of the polymer (16.5 kPa vs. 15 GPa) mechanically isolates each fiber strand and results in nearly discrete stress fields between the fibers (Fig. 2b and Supplementary Movie 2). The resultant stress isolation within individual fibers effectively reduces the overall bending stiffness of the hybrid probe. Interestingly, in the dehydrated state the hydrogel matrix exhibits an order of magnitude greater bending stiffness (53 N m$^{-1}$) (Fig. 2b), which can be attributed to weaker stress isolation effect due to >3000-fold increase in Young's modulus of the dried hydrogel (50 MPa vs. 16.5 kPa) (Supplementary Fig. 8b). To test the validity of the numerical analyses, we further experimentally characterize bending stiffness of the hydrogel hybrid probes within the frequency range of 0.1–10 Hz (i.e., a physiological range for circulation, respiration, and locomotion). The experimentally measured bending stiffness values (252, 77, 4.3, 2.6, and 0.42 N m$^{-1}$ for stainless steel microwires, silica optical fibers, polymer probes, dried hybrid probes, and swollen hybrid probes, respectively with 10 mm length for all cases) are in a quantitative agreement with the values obtained by FEA (Fig. 2c).

The hydration-dependent bending stiffness of the hydrogel hybrid probes renders them adaptive for insertion surgery and long-term biocompatibility within the brain tissue. Compliant probes typically pose a challenge to deterministic implantation into the deep-brain regions, which necessitates the use of additional insertion guides[28,29], temporary coatings[30], or biomimetic-strategies such as a mosquito-inspired approach[31] to assist insertion. While low bending stiffness of the fully swollen hybrid probe similarly results in buckling, which precludes its insertion into the phantom brain (Fig. 2d and Supplementary Movie 3), dried hybrid probe with significantly greater stiffness readily penetrates into the deep brain regions (Fig. 2e and Supplementary Movie 3). Following implantation, the dried hybrid probe converts into the soft swollen probe by quickly absorbing water from the surrounding tissue (the fully swollen state is reached in 10 min) (Fig. 2f and Supplementary Fig. 11a). Note that the swelling of the hydrogel hybrid probe does not result in perturbation in local homeostasis due to the low water uptake volume by the probe (~0.3 μL), and the rate of hydration lower than the rate of cerebrospinal fluid (CSF) perfusion (0.03 μL min$^{-1}$ for hydrogel <0.35 μL min$^{-1}$ for CSF in mice)[32]. This spontaneous change from the stiff insertable state to the soft implanted state provides a promising strategy for implantation of compliant neural probes into the deep brain regions, without demanding additional tools or compromising the biomechanical compatibility.

To investigate mechanical interactions between the hydrogel hybrid probe and the brain tissue following implantation, we develop a FEA model mimicking the experimental and physiological conditions for the brain micromotion[33] (Fig. 2g)

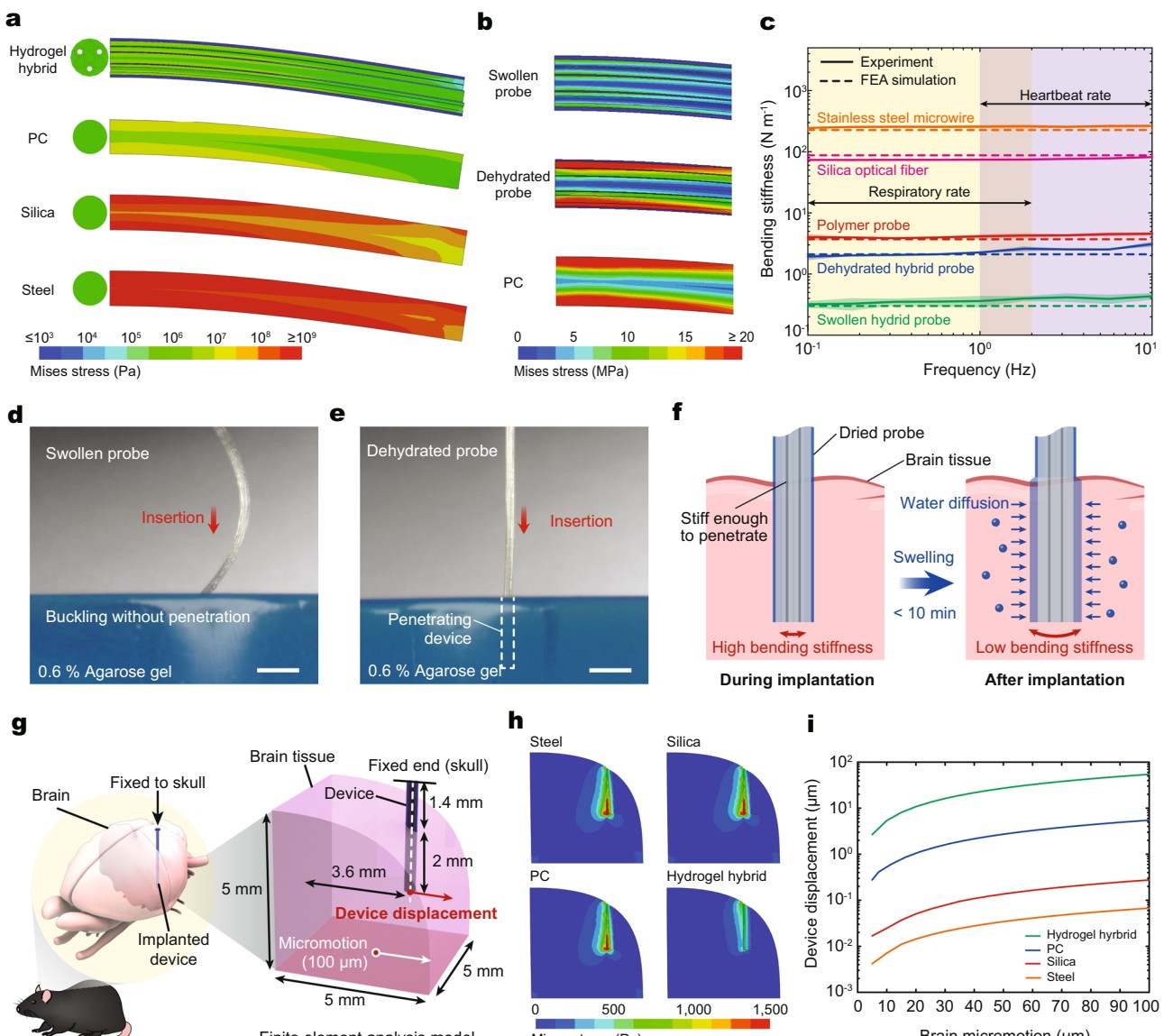

**Fig. 2 Mechanical analysis of hydrogel hybrid probes. a** Mises stress profiles for fibers composed of stainless steel, silica, PC, and for the hydrogel hybrid probes under bending deformation. **b** Expanded view of the FEA results for a PC fiber and a hybrid probe with a fully swollen and a dehydrated hydrogel matrix. The swollen probe exhibits mechanical isolation of individual polymer fibers with discrete stress field development. **c** Experimentally measured bending stiffness for stainless steel, silica, and PC fibers and for the dehydrated and swollen hydrogel hybrid probes in the frequency range of 0.1–10 Hz. **d**, **e** Insertion of the hybrid probe with fully swollen (**d**) and dehydrated (**e**) hydrogel matrix into the phantom brain (0.6% agarose) at a speed of 1 mm s⁻¹. Scale bars: 1 mm. **f** A conceptual illustration of the adaptive bending stiffness of the hydrogel hybrid probes. The dehydrated probes exhibit bending stiffness sufficient for initial insertion. Following implantation, probes absorb water from the surrounding tissue and convert into compliant swollen state. **g** A schematic illustration for the FEA model of the implanted device fixed to the skull during the brain micromotion. **h** Mises stress profiles within the brain tissue for stainless steel, silica, and PC fibers and for the dehydrated and swollen hydrogel hybrid probes at 100 μm lateral micromotion of the brain tissue. **i** The displacement for stainless steel, silica, and PC fibers and for the dehydrated and swollen hydrogel hybrid probes during 0–100 μm lateral micromotion of the brain tissue. Lines and the shaded areas in **c** represent the mean and the standard deviation, respectively ($n = 6$).

(see "Methods" section for details on FEA parameters and setups). In the model, the probe tip is located in the ventral hippocampus (vHPC, coordinate: −3.08 mm AP; 3.6 mm ML; −3.4 mm DV) of the mouse brain, which is selected as the target region for the in vivo experiments in this study. The backend of the probe is fixed to the skull, and the brain tissue repeatedly moves along the lateral direction with 100 μm amplitude (Fig. 2h, Supplementary Fig. 13, and Supplementary Movie 4), and both the lateral and vertical directions with 100 μm amplitudes (Supplementary Fig. 14) to simulate physiological brain micromotion due to circulation, respiration, and locomotion. The numerical analysis indicates that the hydrogel hybrid probes produce lower stress and strain fields within the surrounding tissue during micromotion than the probes composed of steel, silica, and PC (Fig. 2h and Supplementary Figs. 13 and 14). Notably, the hydrogel hybrid probe generates relatively uniform stress and strain fields in the surrounding tissue, while steel, silica, and PC probes result in stress and strain concentration at the probe tips, where sensing and modulation of neural activity take place. Reduced and distributed stress fields observed for hydrogel hybrid probes are anticipated to improve the reliability and biocompatibility of these devices in long-term studies[34].

In addition to tissue damage, relative movements of the neural probes with respect to their target tissues pose a challenge to reliable recording of identifiable neuronal units in extended experiments[33]. The low bending stiffness of the hydrogel hybrid probes results in increased deflection from the anchor point on the skull, and the reduced relative displacement between the device tip and the target brain tissue during micromotion as compared to steel, silica, and PC probes (Fig. 2i and Supplementary Fig. 15). Maintaining relative proximity to the target neurons may permit tracking of their electrophysiological signals with hydrogel hybrid probes over longer periods.

**In vivo electrophysiology and behavioral assays combined with optogenetic interrogation**. To validate the performance of the hydrogel hybrid probes, we apply them in an optogenetic testbed study of the neural projection dynamics in mice (Fig. 3a). We target the excitatory projection from the basolateral amygdala (BLA) to the ventral hippocampus (vHPC), which was shown to mediate anxiety-like behaviors in mice[35]. To gain optogenetic control of BLA axons terminating in the vHPC, we inject an adeno-associated virus (AAV, serotype 5) carrying the gene for a blue-light activated cation channel channelrhodopsin 2 (ChR2) fused to the enhanced yellow fluorescent protein (eYFP) under the excitatory neuronal promoter calmodulin kinase II α-subunit (AAV5-CaMKIIα::ChR2-eYFP, or the corresponding control virus AAV5-CaMKIIα::eYFP) into the BLA and implant hydrogel hybrid devices into the vHPC (Fig. 3b).

We then record electrophysiological signals in the vHPC of freely moving mice in response to optical stimulation (10 Hz, 5 ms pulse width, 10 mW mm$^{-2}$, 1 s stimulation epochs separated by 4 s rest periods) 3 days, 1 week, 2 weeks, and every month between 1–6 months following implantation (Fig. 3c and Supplementary Figs. 16 and 17). Potentials correlated with laser pulses are observed 1–2 weeks after the virus injection, and their amplitudes gradually increase up to 8 weeks, at which point they reach a stable level maintained for at least 4 more months (Fig. 3d). No light-evoked potentials are recorded in mice injected with the control virus (Supplementary Fig. 18). Consistent with prior studies[36], the noise level sharply decreases 2 weeks following the implantation and is maintained for at least 6 months (Fig. 3e). Signal-to-noise ratio (SNR) for the hydrogel hybrid probe gradually increases over the first 4 weeks following the implantation and then plateaus for another 5 months (Fig. 3f). In contrast, the SNR recorded from the polymer probes exhibits gradual decrease and becomes one after 12 weeks following implantation due likely due to a greater foreign body caused by a mechanical mismatch with the surrounding neural tissue[5,12] (Fig. 3f).

The efficacy of the optical neuromodulation with the hydrogel hybrid probes is further corroborated in behavioral experiments. We subject the same cohort of mice to a standard open field test (OFT) (9 min session, 3 min light OFF/ON/OFF epochs) 6 weeks following probe implantation (Fig. 3g). Consistent with previous studies of the BLA-to-vHPC projection circuit[35], optical stimulation (20 Hz, 5 ms pulse width, and 10 mW mm$^{-2}$) of the BLA inputs in the vHPC delivered via the hybrid probes results in reduction of the time spent in the center of the open field by mice expressing ChR2-eYFP as compared to eYFP controls (two-way ANOVA and Bonferroni multiple comparison test, $p = 0.0031$) (Fig. 3h, j, k). Optical stimulation does not yield changes in velocity in either of the groups (Supplementary Fig. 20a).

It has been shown that these changes in anxiety-like behaviors are mediated by the monosynaptic, glutamatergic inputs from the BLA to the vHPC[35]. We recapitulate these findings by using hydrogel hybrid probes to deliver a glutamate receptor antagonist

cocktail (the combination of AMPA (α-amino-3-hydroxy-5-methyl-4-isoxazolepropionic acid) and NMDA (N-methyl-D-aspartate) receptor antagonists, NBQX (2,3-dihydroxy-6-nitro-7-sulfamoyl-benzo[f]quinoxaline-2,3-dione), and AP5 (2R-amino-5-phosphono-pentanoate) through hybrid probes to the vHPC (0.5 µl) during the OFT. These experiments are performed 1 week following the first OFT, and employ the identical optical stimulation paradigm. Consistent with glutamatergic character of the BLA-to-vHPC projection, the NBQX/AP5 cocktail attenuates the optically-induced increase in anxiety-like phenotype in ChR2-expressing mice resulting in loss of significance from the control group ($p > 0.9999$) (Fig. 3i, l, m). Similar to the experiments performed without the injections, the optical stimulation does not induce changes in the velocity (Supplementary Fig. 20b). Note that the unobstructed delivery of pharmacological agents in these experiments via microfluidic channels of the implanted hydrogel hybrid probes, demonstrates microfluidic functionality of these devices at least 8 weeks following implantation.

**Long-term tracking of isolated single-neuron potentials**. We previously found that the polymer fiber-based probes enable tracking of the stable isolated single-neuron potentials up to 3 months following implantation, which was attributed to their flexibility and biochemical inertness[12]. Here, we hypothesized that the significant reduction in stress and strain field levels predicted for hydrogel hybrid probes should result in a substantial extension of the period over, which isolated single neuron action potentials are monitored. To test this hypothesis, we record spontaneous activity in the vHPC of the freely moving mice over a period of 6 months (Fig. 4a–e). Each microelectrode fiber in the hydrogel hybrid probe measured 1 or 2 single unit-potentials (spikes). It should be noted that individual electrodes in each microelectrode fiber were not electrically distinguishable due to collective electrical connectorization procedure, and thus they measured the same pattern. This cross-talk was not a feature of the electrode array design or polymer insulation, which were previously shown sufficient to avoid cross-talk between neighboring channels, but instead was a fabrication choice for this proof-of-concept study. In a representative example, a pair of action potential waveforms likely stemming from two distinct neurons is recorded 3 days, 1 week, 2 weeks, and between 1–6 months with 1 month interval maintains principle component cluster separation as quantified by L-ratio (0.00286) and isolation distance[36] (Fig. 4b). The clusters corresponding to the individual units obtained at different time points are largely overlapping (Fig. 4b), indicating the stability of the waveform shapes and amplitudes[29] (Fig. 4c, d). The different units further maintain their characteristic and statistically distinct interspike intervals (one-way Student's t-test, $p < 0.001$)[36] (Fig. 4e). These results have also been corroborated from the other mice implanted with distinct identically manufactured devices, and we could find the six additional examples of recordings are presented in Supplementary Fig. 21. The number of functional channels and the maximum number of recordable single units per probe considering overlapped patterns (between 2–6) are provided in the Supplementary Table 3. Furthermore, SNR in the recordings acquired with the hydrogel hybrid probes remains stable at least up to 24 weeks following the implantation, whereas the recordings acquired with polymer probes exhibit substantial decrease in SNR 4 weeks following implantation (Supplementary Fig. 21, $p = 0.000578$ and 0.000124 between 4 weeks and 8 or 12 weeks, respectively).

**Assessment of the foreign body response to the hydrogel hybrid probes**. To assess the foreign body response to the

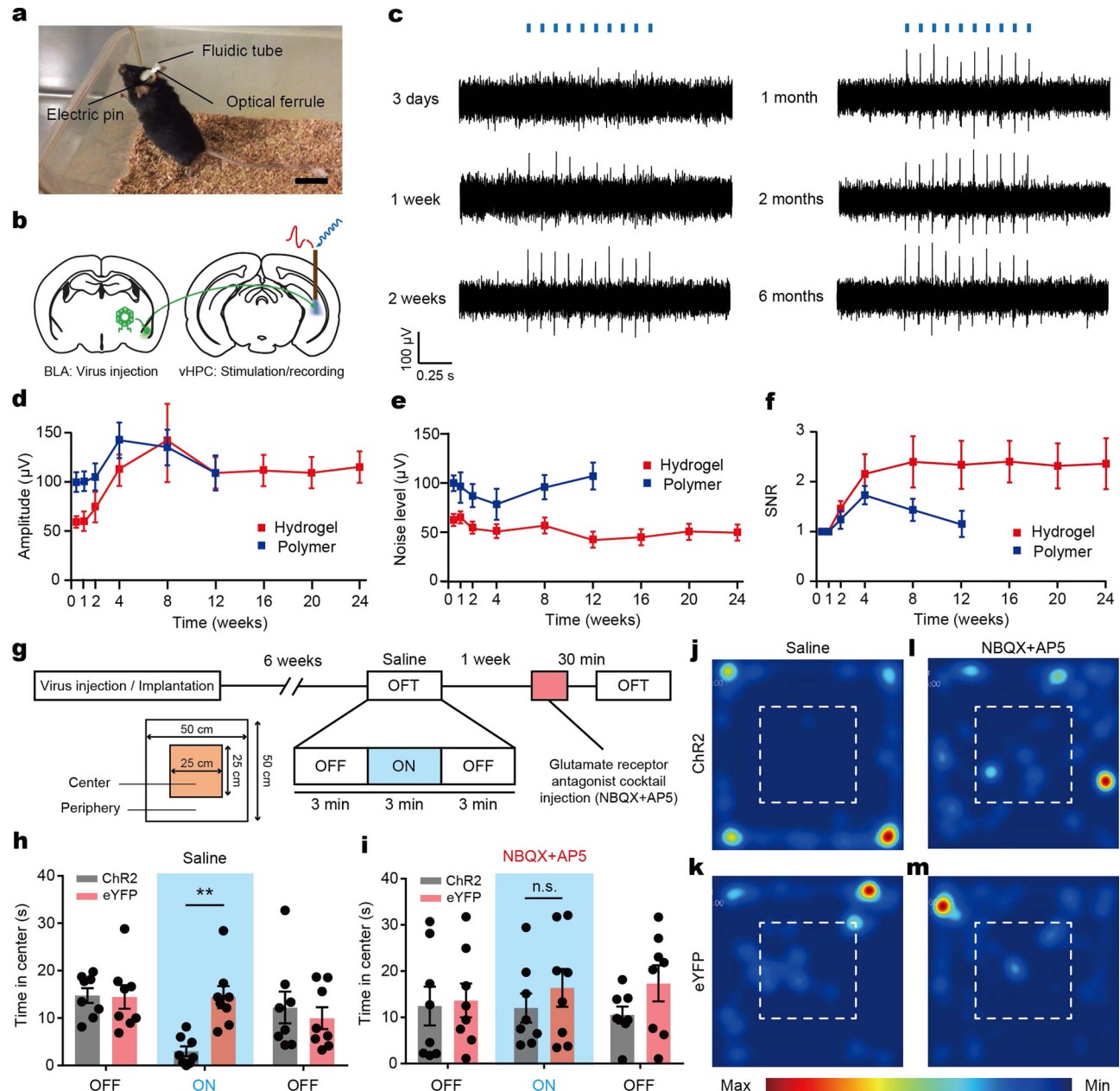

**Fig. 3 Optogenetic interrogation by hydrogel hybrid probes. a** A photograph of a mouse implanted with the hydrogel hybrid probe. Scale bar: 2 cm. **b** An illustration of optogenetic modulation and electrophysiological recording in the BLA-to-vHPC projection circuit. **c** Representative electrophysiological recordings in the vHPC during optical stimulation (10 Hz, 10 mW mm$^{-2}$, 5 ms pulse width) using the hydrogel hybrid probes 3 days, 1–2 weeks, and 1–6 months following implantation and transfection with AAV5-CaMKIIα::ChR2-eYFP. **d–f** Amplitude of optically-evoked potentials (**d**) and background noise (**e**), and signal-to-noise ratio (SNR) (**f**) in the opto-electrophysiological experiments recorded from the hydrogel hybrid probes (red) and the polymer probes (blue) at various time point ($n = 8$ for each). **g** A summary of the behavioral assays performed in this study. Open field tests (OFT) are performed (3 min light OFF/ON/OFF epoch) 6 weeks following the injection of AAV5-CaMKIIα::ChR2-eYFP or AAV5-CaMKIIα::eYFP. Time spent in the center region of the OFT arena during optical stimulation (20 Hz, 10 mW mm$^{-2}$, 5 ms pulse width) is analyzed for trials conducted with the injections of saline solution or glutamate receptor antagonist cocktail (NBQX + AP5). Optical stimuli and drug injections are delivered into the vHPC via the hydrogel hybrid probes. **h** Time spent in the center in the absence or presence of optical stimulation following injection of saline solution (two-way ANOVA and Bonferroni multiple comparison test, $p = 0.0031$) **i** Time spent in the center in the absence or presence of optical stimulation following injection glutamate receptor antagonist cocktail (two-way ANOVA and Bonferroni multiple comparison test, $p > 0.9999$). **j–m** Representative heat map images tracing the position of mice transfected with ChR2 (**j**, **l**) and eYFP (**k**, **m**) control with injection of saline solution (**j**, **k**) and glutamate receptor antagonist cocktail (**l**, **m**). Dashed line indicates the center region of the open field arena. Values in **d–f**, **h**, **i** represent the mean and the standard deviation ($n = 8$).

hydrogel hybrid probes, we characterize the expression of markers of glial scarring: glial fibrillary acidic protein (GFAP) for astrocytes; ionized calcium-binding adapter molecule 1 (Iba1) for microglia; and cluster of differentiation 68 (CD68) for activated macrophages; as well as the presence of immunoglobulin G (IgG),

a marker of the breach of the blood-brain barrier in the vicinity of these devices 3 days, 1 week, 1 month, 3 months, and 6 months following the implantation surgeries (Fig. 4f–i, Supplementary Fig. 23, and Supplementary Table 5). Immunohistochemistry in coronal slices is used to compare the responses to the hydrogel

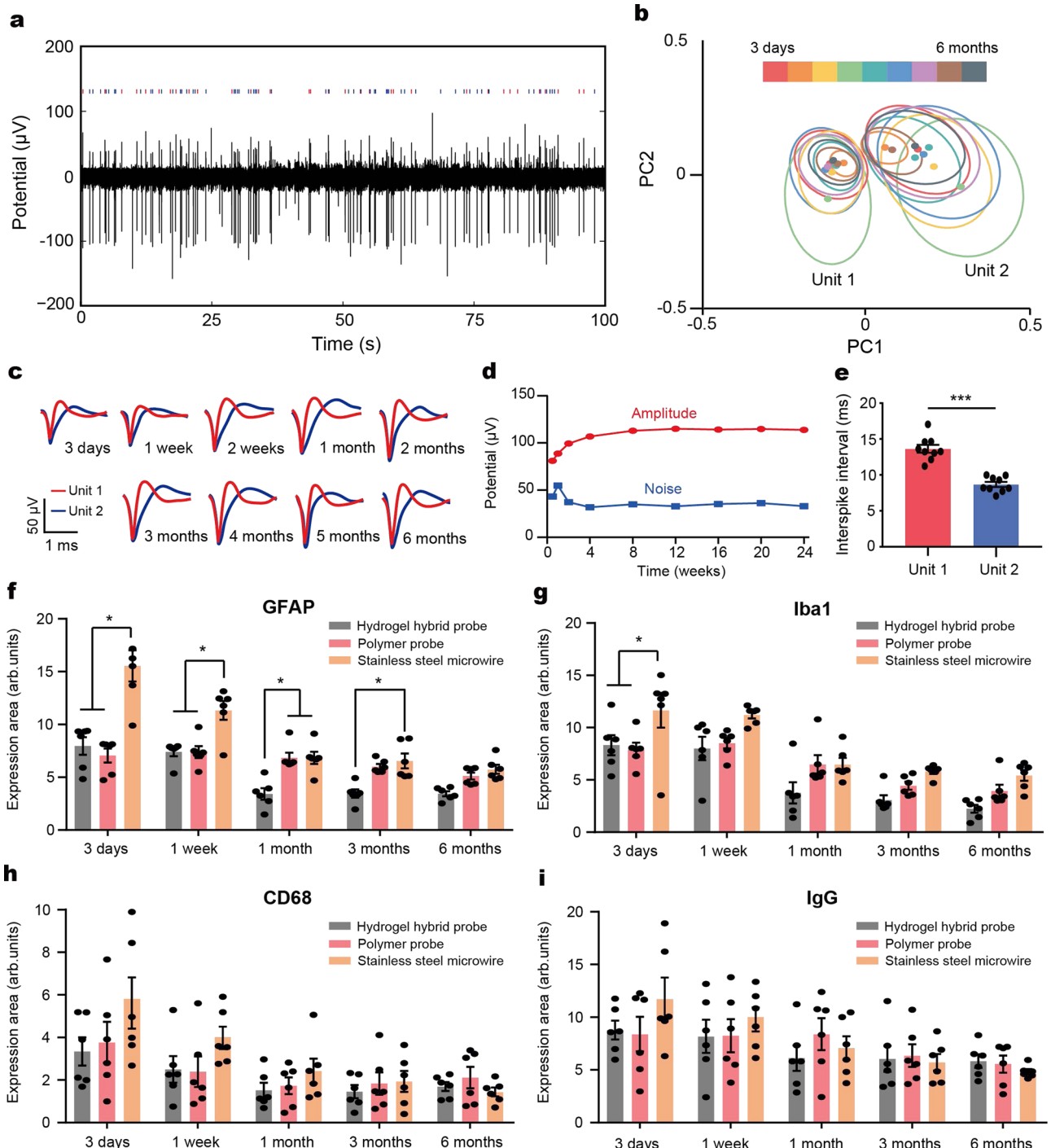

**Fig. 4 Long-term investigation of single-unit potentials and foreign body responses. a** Representative spontaneous electrophysiological activity recorded in the vHPC using a hydrogel hybrid probe 1 month following implantation. **b** Principle component analysis of the single-unit potentials (spikes) recorded 3 days, 1 week, 2 weeks, and 1–6 months following implantation. Closed circles indicate centers and ellipses represent two standard deviations ($2\sigma$) contours of the principle component distributions. **c** Average spike waveforms recorded over time corresponding to clusters in **b**. **d** Average amplitude (red) and noise level (blue) of spikes in **c** recorded over time. **e** Average inter-spike intervals for each unit. Significance is confirmed by one-way Student's $t$-test (***$p < 0.001$). **f–i** Average immunofluorescence area quantifying the presence of GFAP (**f**), Iba1 (**g**), CD68 (**h**), and IgG (**i**) in the vicinity of the hydrogel hybrid probes, polymer probes, and stainless steel microwires for 3 days, 1 week, 1 month, 3 months, and 6 months following implantation, respectively. Values in **f–i** represent the mean and the standard deviation ($n = 6$ for each device and time point, Kruskal–Wallis test and Dun–Sidak multiple comparison test (*$p < 0.05$).

hybrid probes, polymer probes, and stainless steel microwires with similar dimensions.

Initially the tissue reaction to the implanted probes is dominated by the insertion effects (e.g., acute tissue damage during implantation), whereas in the long-term the tissue response is predominantly defined by the chronic tissue-device interactions such as micromotion[34] (Supplementary Fig. 24). Hence, 3 days and 1 week following implantation the brain tissue

surrounding hydrogel hybrid probes exhibits significantly lower biomarker expression compared to that surrounding the steel microwires, while the tissue response is similar to that observed for the polymer probes due to the similar initial bending stiffness of the dehydrated hydrogel probes. At longer post-implantation time points (1, 3, and 6 months), the hydrogel hybrid probes elicit substantially reduced foreign body response as compared to both polymer probes and steel microwires (Fig. 4f–i), which is highly correlated with the SNR of the recorded signals (Supplementary Fig. 24).

## Discussion

In this study, we develop adaptive, biocompatible, and multi-functional hydrogel hybrid devices with extended long-term stability for opto-electrophysiological studies of deep brain circuits in freely moving subjects. The intimate integration of thermally drawn polymer fibers within a hydrogel matrix addresses chemo-mechanical mismatch, between the synthetic devices and neural tissues and affords multiple functions. The soft hydrogel matrix effectively decouples the stress fields in individual polymer fibers within the hybrid probes during bending, leading to a significantly lower bending stiffness of these devices than that of solid polymer, glass, or metal probes with identical dimensions. As a result, the hybrid probes induce substantially lower stress and strain in the surrounding neural tissues during the micro-motion of the brain with respect to the skull. In addition, the three orders of magnitude difference between the dried and fully-swollen hydrogels' elastic moduli results in adaptive properties, which allow for straightforward insertion into deep brain structures while minimizing the long-term biomechanical mismatch with the neural tissue. This combination of multifunctionality, favorable tissue-device interactions, and adaptive properties may offer a promising direction for future development of other hydrogel-based devices including wearable and implantable bio-marker sensors, impacting innovations in body-machine interfaces.

Although the hydrogel hybrid probes offer functional advantages in tissue response, there are several points of further improvement left for the future development and applications. First, the complexity, size, and recording capacity of the micro-electrode fibers are limited by the back-end electrical connectorization and resultant overlap in signal patterns from adjacent electrodes in each microelectrode fiber. An improved approach to establish high-resolution connections between functional fibers and back-end interfaces would enhance the throughput of recorded units. Also, while here we focus on mechanical aspects of hydrogel–hybrid probe biocompatibility and longevity, this platform may offer additional strategies for improving long-term performance. For instance, incorporation of biofunctional substances such as growth factors, neuroattractants, and cell-adhesion enhancers within the hydrogel matrix may further offer biochemical advantages to promote the intimate integration of the tissue with the probe elements[37–39]. Alternatively, biodegradable hydrogel or polymer bodies[40,41] may permit the implantation of miniaturized functional elements with dimensions <10 μm. Such floating components may minimize disruption of glial and neural networks and negate the physiological effects of the implantation[8,9,42,43]. Taken together, the advances demonstrated in this work may open opportunities for future studies of neurobiological phenomena requiring wider experimental time windows (>6 months) including development and aging, and fundamental understanding of progressive neurological disorders such as Parkinson's disease and Alzheimer's disease.

## Methods

**Multifunctional fiber fabrication**. The preforms, macroscale templates for the thermal drawing process were first fabricated. To build the optical waveguide preform, cyclic olefin copolymer (COC; TOPAS) and additional polycarbonate (PC; McMaster-Carr) sheets were rolled onto a PC rods (16 mm diameter), and then the whole structure was consolidated under vacuum at 190 °C. The preforms for the multi-electrodes were fabricated by drilling the holes (6.35 mm diameter) into poly(etherimide) (PEI) rods (31.75 mm diameter, McMaster-Carr), and by inserting seven tin rods (Sn; 6 mm diameter, Alfa Aesar) inside of the holes. Additional poly(phenylsulfone) (PPSU) sheets were rolled on the structure to provide the sacrificial layers. For the fluidic channel preform, one large hole was made by drilling to the same PEI rod with above. Both were stored in the vacuum oven at 250 °C. All preforms were then thermally drawn using a custom-built drawing tower. The temperature for the drawing process was 240 °C for the waveguide, and 315 °C for multi-electrodes and fluidic channels. The multi-electrode fibers were drawn one more for the smaller features, and PPSU cylinder (31.75 mm diameter) was used as sacrificial layer for the second drawing.

**Assembly of multifunctional fibers**. The sacrificial PC layer of waveguides were etched by immersion into dichloromethane (Sigma) for 1 min, and PPSU layer of electrodes were removed by dipping the fiber in tetrahydrofuran (Sigma) for 25 min. All etched fibers were inserted into the guide fixture made of microlaser-machined polyimide sheet (250 μm thickness with seven 100 μm diameter holes, A-laser), and glued on it with 5-min epoxy (Devcon). After the stabilization, each fiber was connectorized appropriately to the external parts. For the optical coupling, the waveguides were inserted into 6.5 mm long, 1.25 mm diameter zirconia ferrule (Thorlabs), and affixed by optical epoxy. After the overnight incubation, the ferrule edge was polished using Thorlabs fiber polishing kit for better optical transmission. Sn electrodes were connected to the pin connector (Digi-Key) with following procedures. The silver paint (SPI Supplies) was first applied on the tip of the electrode fiber. Copper wires were wrapped on the tip after 20 min of that, which allowed the enough evaporation of the paint thinner. The pins of electrical connectors were soldered to the copper wires and the insulated stainless steel ground wires. Microfluidic channels were inserted in the ethylene vinyl acetate tubing (0.5 mm inner diameter, McMaster-Carr), and the gap between the channel and tubing was filled with 5 min epoxy. Additional epoxy was applied to the whole parts to provide the mechanical stability and electrical insulation of the devices.

**Surface functionalization of fiber assembly**. Before surface functionalization, the multifunctional fiber assembly was thoroughly cleaned by isopropyl alcohol and dried with nitrogen flow. To functionalize primary amine to polymer surface, the cleaned fiber assembly was incubated in 10 v/v% aqueous hexamethyldiamine (HMDA; Sigma) solution for 24 h at room temperature. After incubation, the fiber assembly was thoroughly cleaned with deionized water and dried with nitrogen flow. Immediately after the HMDA treatment, the polymer surface was further grafted with alginate via carbodiimide chemi. To prepare alginate grafting solution, 0.1 g of sodium alginate (Sigma), 50 mg of 1-Ethyl-3-(3-dimethylaminoprophyl) carbodiimide (EDC; Sigma), and 15 mg of N-Hydroxysulfosuccinimide (Sulfo-NHS; Sigma) were added per 10 ml MES buffer (Sigma). To complete alginate grafting, the primary amine functionalized fiber assembly was incubated in the alginate grafting solution for 12 h at room temperature. After incubation, the fiber assembly was thoroughly cleaned with deionized water and dried with nitrogen flow. Functionalized fiber assembly was kept in nitrogen environment before further processing.

**Detection of surface functionalized amine and alginate**. The presence of surface functionalized primary amine groups was quantified via Coomassie brilliant blue (CBB) colorimetric assay. To prepare CBB solution, 500 mg of CBB (Sigma) was added into the mixture of 10 ml methanol and 5 ml glacial acetic acid (Sigma). Then, deionized water was added into the CBB solution until the final volume reached 100 ml. To prepare washing solution, 10 ml methanol and 5 ml glacial acetic acid were added into 85 ml deionized water. To detect the presence of surface functionalized primary amines, samples were immersed into the diluted CBB solution and incubated for 1 h at room temperature with mild agitation. After incubation, the samples were cleaned with the washing solution three times. The presence of primary amine groups was determined by checking blue tint of CBB dye molecules bound to the surface of the samples. The presence of surface grafted alginate was quantified via fluorescein covalently coupled to alginate. To prepare fluorescein coupling solution, 50 mg of EDC, 15 mg of Sulfo-NHS, and 20 mg of 6-aminofluorescein (Sigma) were added per 10 ml MES buffer. To detect the presence of surface grafted alginate, samples were immersed into the fluorescein coupling solution and incubated for 12 h at room temperature. After incubation, the samples were cleaned with deionized water three times. The presence of surface grafted alginate was determined by checking green fluorescence under a fluorescent microscope (Eclipse LV 100ND, Nikon).

**Integration of hydrogel matrix with fiber assembly**. Pre-gel solution of PAAm-Alg hydrogel was prepared by adding 16 w/w% of acrylamide (Sigma), 2 w/w% of

sodium alginate, 2 w/w% of glucose (Sigma), 0.02 w/w% of *N,N'*-methylenebis (acrylamide) (Sigma), 0.2 w/w% of 2-hydroxy-4'-(2-hydroxyethoxy)-2-methyl-propiophenone (Sigma), 0.02 w/w% glucose oxidase (Sigma), and 50 mM calcium sulfate (Sigma) in deionized water. To integrate hydrogel matrix with the surface functionalized fiber assembly, the fiber assembly was dip-coated with the hydrogel pre-gel solution at 1 mm s$^{-1}$ pulling speed under nitrogen flow. Superfluous hydrogel pre-gel solution on the fiber assembly was further removed by the scraper made of micro laser-machined polyimide sheet (250 μm thickness, A-laser) to ensure the uniform thickness of hydrogel matrix (thickness: 25 μm). Immediately after the dip-coating process, the hydrogel matrix was polymerized/crosslinked by incubating in a humid UV chamber (CL-1000, UVP) filled with nitrogen for 30 min. During dip-coating and UV curing processes, glucose and glucose oxidase added in the pre-gel solution served as an oxygen scavenger to prevent oxygen inhibition effect on polymerization and crosslinking of acrylamide monomers[17]. After UV polymerization/crosslinking, the prepared hydrogel hybrid probes were kept in a large volume of Dulbecco's phosphate buffered saline (DPBS) at least for 3 days to leach out unreacted regents. The tip of hydrogel hybrid probe was cut by a sterile razor blade before implantation in order to ensure exposure of the functional fibers.

**Preparation of various hydrogels.** Poly(vinyl alcohol) (PVA) hydrogels were prepared by adding 30 μL of 25% glutaraldehyde solution (Sigma) and 10 μL of 37% hydrochloric acid (HCl) solution per 1 mL of aqueous 10 w/w% PVA (Mw 130,000 with 99+% hydrolyzed; Sigma) solution. Alginate hydrogels were prepared by adding 2 w/w% sodium alginate, 5 mM adipic acid dihydrazide (Sigma), 5 mM EDC, and 5 mM Sulfo-NHS in MES buffer. Poly(ethylene glycol) diacrylate-alginate (PEGDA-Alg) hydrogels were prepared by adding 20 w/w% PEGDA (Mw 20,000; Sigma), 2 w/w% sodium alginate, 0.2 w/w% of 2-hydroxy-4'-(2-hydroxyethoxy)-2-methylpropiophenone, and 50 mM calcium sulfate in deionized water. PVA and alginate hydrogels were crosslinked in room temperature for 6 h. PEGDA-Alg hydrogels were crosslinked by incubating in a humid UV chamber (CL-1000, UVP) filled with nitrogen for 30 min. All hydrogels were fully swollen in DPBS before mechanical characterizations.

**Mechanical characterization of hydrogels.** All mechanical characterizations of hydrogels were performed in ambient air at room temperature with a mechanical testing machine (Z2.5, Zwick/Roell). All hydrogel samples maintained consistent mechanical properties over the time of the tests (~ a few minutes), during which the effect of dehydration was not significant. Tensile tests of swollen and dried hydrogels were performed with the samples (length × width × thickness = 20 mm × 60 mm × 2 mm) at 0.05 s$^{-1}$ strain rate. The nominal stress vs. stretch curves of the swollen hydrogels were fitted with the incompressible neo-Hookean model to obtain shear moduli for each hydrogel. Young's modulus of the swollen PAAm-Alg hydrogel was calculated as three times of the shear modulus, following the incompressible neo-Hookean model. The nominal stress vs. nominal strain curve for the dried PAAm-Alg hydrogel was linearly fitted at 0.2% strain to obtain Young's modulus. Fracture toughness of hydrogels was obtained from tensile tests of notched and unnotched samples, according to the previously reported method. Interfacial toughness between hydrogels and PC substrate was obtained by 90°-peel tests according to the previously reported method. Briefly, the hydrogel samples (length × width × thickness = 50 mm × 10 mm × 2 mm) on PC substrates were peeled at a constant peeling speed of 50 mm min$^{-1}$ using 90°-peel fixture (G50, Test Resources). The measured peeling force reached a plateau with slight oscillations as the peeling process entered steady-state. The interfacial toughness was determined by dividing the plateau force by the width of the sample. Note that the PC substrates used for 90°-peel tests were surface functionalized with primary amine for PVA hydrogels and with alginate for alginate, PEGDA-Alg, and PAAm-Alg hydrogels based on the abovementioned method.

**HPLC characterization of PAAm-Alg hydrogel.** The residual monomer contents of the PAAm-Alg hydrogel were analyzed based on an analytical HPLC (Model 1100, Agilent). DPBS was used as a mobile phase, extractant, and media for an acrylamide monomer standard solution following the previously reported protocol[24]. Before HPLC analysis, the prepared PAAm-Alg hydrogel was kept in a large volume of DPBS for 3 days, following the same procedure for the hydrogel hybrid probe. To extract the residual monomer from the washed PAAm-Alg hydrogel, 100 mg of the PAAm-Alg hydrogel was incubated in 20 mL of the extractant for 24 h with stirring. After the extraction, the solution was filtered with a sterile 0.2 μm syringe filter and injected into the HPLC system for analysis. The concentration of the residual monomer in the PAAm-Alg hydrogel was determined based on the calibration curve obtained from the standard solution diluted with the mobile phase to varying monomer concentrations.

**Characterization of hydrogel hybrid probes.** To measure the optical property of the hydrogel hybrid probes, a diode-pumped solid state (DPSS) laser (50 mW maximum output, wavelength $\lambda$ = 473 nm, Laserglow) was coupled via optical cable and a ferrule connected to an optical waveguide. The light output from the tip of the device were measured by a photodetector (S121C, 400–1100 nm, 500 mW, Thorlabs) attached to a power meter (PM100D, Thorlabs). Measurements were

conducted for the fibers length between 0–2 cm (0.5 cm interval), bending angles 0°, 90°, and 180° with a radius of curvature 1 mm. To characterize the electrical property, electrochemical impedance spectroscopy (EIS) was performed to obtain the impedance at different frequencies (100 Hz–100 kHz) at no and 90° of bending deformation. Microfluidic channels were characterized by connecting the external tubing with a standard injection apparatus (NanoFil syringe & UMP-3 Syringe pump, Word Precision Instruments). Return rate was calculated with the infusion of 10 μl water at the rate of 0.1–10 μl min$^{-1}$. Bending stiffness of the devices were measured by using a dynamic mechanical analyzer (Q800, TA Instruments) with 50 μm deflection amplitude within the frequency range of 0.1–10 Hz (i.e., physiologically relevant frequency range). To test insertion capability of the devices, the hybrid probes with fully swollen and dried hydrogel matrix were inserted into the brain phantom (0.6% agarose hydrogel) with 90° insertion angle at 1 mm s$^{-1}$ insertion speed. To aid visual representation, the agarose hydrogel was dyed with blue food dye (McCormick).

**Numerical analysis of bending stiffness.** The 3D finite element models were established to evaluate the bending stiffness for various cylindrically shaped probes with the same dimension (0.334 mm cross-section diameter and 3.4 mm length). Different mechanical properties (i.e., Young's modulus and Poisson's ratio) were used to represent different materials in each probe (Supplementary Table 4). For the hydrogel hybrid probe, the same structural design in Fig. 1c was adopted. The FEA simulations were implemented by commercial software package ABAQUS 2016 with 8-node 3D brick element C3D8. In the simulations, one end of the probe was fixed, and another end was set free to deflect. Upon a small point load applied *F* at the free end of the probe pointing perpendicular to the probe axis direction, the probe was bent with the deflection *u* and the bending stiffness *K* was computed by the relationship $K = Fu^{-1}$. The simulation results were also compared with the analytical solution for bending stiffness $K_{theory} = (3\pi Ed^4)(64\ L^3)^{-1}$, which showed a good agreement with the simulation results.

**Numerical analysis of brain micromotions.** The 3D finite element models were established to evaluate the interactions between the implanted probes and the brain tissue during micromotions. Both swollen hydrogel and brain tissue were represented by incompressible neo-Hookean material with shear moduli of 3 kPa and 5.5 kPa, respectively, and implemented with 8-node, hybrid, reduced integration 3D brick element C3D8RH in ABAQUS 2016. Other materials were implemented with 8-node 3D brick element C3D8 with the corresponding mechanical properties (Supplementary Table 4). Prior to the micromotion, the top end of the implanted probe was fixed while another end set to free for deflection. To simulate the brain micromotion, the bottom surface of the brain tissue was loaded laterally with a 100 μm displacement (Fig. 2g), or both laterally and vertically with 100 μm displacements for each direction (Supplementary Fig. 14). A surface-to-surface contact interaction between the probe and the brain tissue was modeled as friction with a friction coefficient of 0.3.

**Implantation of hybrid devices into mouse brain.** All animal procedures were approved by the MIT Committee on Animal Care and carried out in accordance with the National Institutes of Health Guide for the Care and Use of Laboratory Animals. Male C57BL/6 mice aged 6–8 weeks (Jackson Laboratory) were used for the study, and all surgeries were conducted under aseptic conditions. Mice were anaesthetized by isoflurane gas (1.5% in oxygen) with nose cone, and the head was fixed to the stereotaxic apparatus (David Kopf Instrument). Head skin was incised to expose the skull with appropriate preparation of betadine and ethanol scrub. All the coordinates of the target regions were made with respect to the Mouse Brain Atlas. Adeno-associated viruses serotype 5 (AAV5) carrying CaMKIIα::hChR2 (H134R)-eYFP and CaMKIIα::eYFP plasmids were purchased from University of North Carolina Vector Core at concentrations of 2 × 10$^{12}$ particles ml$^{-1}$ and 3 × 10$^{12}$ particles ml$^{-1}$, respectively. The 0.5 μl virus was injected into the basolateral amygdala (BLA, −1.6 mm AP; 3.3 mm ML; −4.9 mm DV) at 100 nl min$^{-1}$ rate with the nanofill syringe (33-gauge needle) and the micropositioner (UMP-3, Word Precision Instruments). After the injection, syringes were raised 0.1 mm to accommodate the injected volume. The hydrogel hybrid probes were then implanted into the ventral hippocampus (vHPC, −3.08 mm AP; 3.65 mm ML; −3.4 mm DV). The miniature screws (McMaster-Carr) were fixed to the skull, and stainless steel ground wire (Goodfellow) connected to a pin connector was soldered on it. The devices were firmly mounted with a layer of medical adhesive (C&B Metabond, Parkell) and a dental cement (Jet-Set 4, Lang Dental). Pre-operative analgesia with subcutaneous injection of Buprenorphine (slow release at 1 mg kg$^{-1}$) was performed. Following the recovery, mice were housed and maintained at 22 °C, 12 h light/dark cycle, 50% humidity with provided the food and water ad libitum.

**In vivo electrophysiology.** For electrophysiological recording, the pin connectors of the hydrogel hybrid probes were connected to a recording system (PZ2-32 head stage and RZ5D system, Tucker Davis Technologies). The DPSS laser was used for the optical illumination through the fiber-ferrule connecting system. Five microseconds width and 10 Hz frequency of optical pulse trains (1 s stimulation and 4 s rest epochs) were used to get the optically-evoked potentials. At the same time, signals were recorded with 50 kHz sampling frequency and filtered in the range of

0.3–5 kHz. Spikes were detected by a commercial software (Offline Sorter, Plexon) and clustered in the first and second principal components (PC1–PC2) plane using k-means clustering. Amplitude and noise level of evoked and spontaneous potentials were calculated with MATLAB and the amplitude of signal, when no activities triggered assumed to be same with noise level. In the signal analysis of the device electrical performance (amplitude, noise, and SNR), eight animals were used (one probe implanted per animal) and the average value of one electrode from different probe was used. Multiple electrodes in a single device were not considered to exclude the device-dependency.

**Behavioral assays**. The animals were subjected to a standard open field test (OFT) (50 cm × 50 cm chamber made of white plastic, which was virtually divided into the center (25 cm × 25 cm) and the periphery) 6 weeks following the virus injection and the device implantation. All the tests were performed with randomly allocated experimental groups. The viral injected wild type mice (AAV5-CaMKIIα::hChR2 (H134R)-eYFP as experimental groups or AAV5-CaMKIIα::eYFP as control groups) into the BLA unilaterally were brought into the room 2 h before the starting the experiment, and placed in the center of the open field for 10 min to recover from the handling stress. Mice were then exposed to the optical stimulation (λ = 473 nm, 20 Hz, ~10 mW mm$^{-2}$ at the tip of the fiber) into the vHPC through the implanted devices with three 3-min epochs (OFF-ON-OFF stimulation, 9 min total). For the pharmacological experiments, the glutamate receptor antagonist cocktail (the combination of AMPA and NMDA receptor antagonists, 22 μM of 2,3-dihydroxy-6-nitro-7-sulfamoyl-benzo[f]quinoxaline-2,3-dione (NBQX) and 38 μM of (2R)-amino-5-phosphono-pentanoate (AP5) in saline solution) was injected to the vHPC 1 week after. OFT were then repeated after waiting 30 min following the injection. Tracking of mouse behavior was done using the EthoVision XT (Noldus).

**Immunohistochemistry analysis of foreign-body response**. Hydrogel hybrid probes, custom-drawn polycarbonate fibers (300 μm in diameter), and stainless steel microwires (300 μm in diameter, Goodfellow) were implanted into vHPC of wild type mice ($n = 6$ per group). The expression of targeted proteins (GFAP, Iba1, CD68, and IgG) were analyzed after the immunohistochemistry at the time points of 3 days, 1 week, 1 month, 3 months, and 6 months ($n = 6$ per time point). For the histology, first fetal plus solution (100 mg kg$^{-1}$ in saline) was IP injected to anesthetize the mice. Solution of 4% paraformaldehyde (PFA) in PBS (pH 7.3) was then pumped transcardially for the perfusion, and extracted brains were fixed in 4% PFA solution overnight. The brains were sliced to 40 μm-thick coronal sections with a vibrating blade microtome (VT1000S, Leica), and washed by the solution with 0.3% Triton X-100 and 3% normal donkey serum in PBS for 1 h. This was followed by overnight incubation in a solution of primary antibodies (goat anti-GFAP 1:1000 (ab53554, Abcam); goat anti-Iba1 1:500 (ab107159, Abcam); rabbit anti-CD68 1:250 (ab125212, Abcam); donkey anti-mouse IgG conjugated to Alexa Fluor 488 1:1000 (A21202, ThermoFisher Scientific) with 3% normal donkey serum in PBS at 4 °C. The sections were washed three times with PBS for 30 min, and then stained with secondary antibodies (donkey anti-goat labeled with Alexa Fluor 633 1:1000 (A21082, ThermoFisher Scientific) or donkey anti-rabbit labeled with Alex Flour 488 1:1000 (A21206, ThermoFisher Scientific)) for 2 h at room temperature. After washing three times again, slices were mounted on glass microscope slides by a Vectashield antifade mounting medium with 4′6-diamidino-2-phenylindole (DAPI; Vector Laboratories). A laser scanning confocal microscope (Fluoview FV 1000, Olympus) with 20× objective (oil, NA = 0.85) were used for image acquisition. ImageJ (ver. 1.8.0_172) was used to quantify the area of express antibodies. All the images were transformed to the 8-bit binary images, and the expression area is calculated with normalized analysis. All analyses were blinded with respect to the experimental conditions.

**Statistical analysis**. GraphPad Prism 7 (GraphPad) software was used to assess the statistical significance of all comparison studies in this work. Sample sizes were determined on the basis of the previous research conducted in the same brain circuits. Normality of data distribution was tested via Shapiro–Wilk test. For the analysis of the behavioral data that showed normal distribution, two-way ANOVA followed by Bonferroni's multiple comparison test and Tukey's multiple comparison test were conducted with threshold of *$p < 0.05$, **$p < 0.01$, ***$p < 0.001$. For immunohistochemistry experiments, where data distributions were found to be non-normal, Kruskal–Wallis test followed by Bonferroni's correction and Dunn–Sidak multiple comparison test were conducted with threshold of *$p < 0.05$. For the fiber characterization, paired two-sided Student's $t$-tests were used, and significance threshold was placed at *$p < 0.05$, **$p < 0.01$. Error bars represent standard deviation.

**Reporting summary**. Further information on experimental design is available in the Nature Research Reporting Summary linked to this paper.

## Data availability
All data supporting the findings of this study are available within the Article and its Supplementary Information. The data necessary to reproduce the figures within this manuscript are provided in the Source Data archive file. Additional raw data generated in this study are available from the corresponding author upon reasonable request. Source data are provided with this paper.

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

## Acknowledgements

This work was supported in part by the National Institute of Neurological Disorders and Stroke (5R01NS086804, P.A.), National Institutes of Health (1R01HL153857-01, X.Z.), National Science Foundation (NSF) Center for Materials Science and Engineering (DMR-1419807, P.A. and Y.F.), NSF Center for Neurotechnology (EEC-1028725, P.A.), the McGovern Institute for Brain Research at MIT (P.A.), NSF (EFMA-1935291, X.Z.), and the U.S. Army Research Office through the Institute for Soldier Nanotechnologies at MIT (W911NF-13-D-0001) (X.Z. and Y.F.). S.P. is currently supported by the Basic Science Research Program through National Research Foundation of Korea (NRF-2020R1C1C1007589), Korea Medical Device Development Fund grant funded by the Korea government (the Ministry of Science and ICT, the Ministry of Trade, Industry and Energy, the Ministry of Health & Welfare, the Ministry of Food and Drug Safety) (NTIS Number: 9991006805), National R&D Program through the National Research Foundation of Korea (NRF) funded by Ministry of Science and ICT (2021M3H4A1A03048648/2021M3F3A2A01037365), Smart Project Program through KAIST-Khalifa Joint Research Center (KK-JRC), KAIST College of Engineering Global Initiative Convergence Research Program. KAIST Post-AI Research Project S.P. and H.Y. were supported by Samsung Scholarship during their graduate studies.

## Author contributions

S.P., H.Y., X.Z., and P.A. designed the study. Y.F. facilitated the fiber design. S.P., E.W.W., and J.K. fabricated and characterized multifunctional fibers. H.Y. prepared and integrated hydrogel matrix for the hybrid probes. H.Y. and R.Z. conducted mechanical characterization and finite element analysis. S.P. performed in vivo electrophysiology and immunohistochemistry. G.B.C. facilitated the design and analysis of behavioral experiments. S.P. and Y.S.Y. conducted behavioral tests. All of the authors contributed to the writing the manuscript.

## Competing interests

The authors declare no competing interests.
