## [Peer Review File · Nature Communications]

REVIEWER COMMENTS

Reviewer #1 (Remarks to the Author):

In this revised version, Park et al. improved the text and the figures. The authors addressed some of the points raised in the previous review. The technology described in the paper shows some more convincing potentials for long-term unit recording for neuroscience applications compared to other (polymer) probes (Supplementary Fig. 19, 22). Hydrogel probes also lead to a lower amount of astrocyte and microglia activation (Fig. 4 and supplementary Fig. 23) compared to stainless steel probes. There are, however, some important points that remain to be addressed in the text. Specifically:

1) It is still unclear to this referee how many channels per probe remain functional over implantation time and how many units can be simultaneously recorded with one probe (see also comment #5 of previous Referee#2). This needs to be explicitly stated in the results and commented in the discussion. It seems that the maximal number of 2 units was recorded from one probe per animal out of 21 recording sites.

2) Supplementary figure 16: the electrical signals triggered by optogenetic stimulation fall within the illumination window in panel b, but not in panel a. There seems to be a mistake there.

3) Supplementary figure 20: the data in panel a, fourth bar from the left (Saline, light ON, eYFP), appear to be the same data shown in panel b, fourth bar from the left (NBQX+AP5, light ON, eYFP). Yet, the two experimental conditions are different. Please comment/amend.

4) In the statistics section, it is stated that error bars in all graphs represent s.e.m., but in figure legends (e.g., Fig. 1 h-j) it is reported that error bars represent SD.

5) Lines 269-270: the decrease in SNR 4 weeks after implantation in polymer probes needs to be statistically quantified.

6) A normality test must be run before choosing the statistical test to be used (parametric vs non parametric), rather than assuming normality based on previous work.

7) Line 265 and lines 234-235: please specify the statistical test used.

8) Fig. 11: the text reports that the electrical properties of the hybrid device are not significantly affected (either increased or decreased, both directions) by bending. However, the test is one sided (as stated in the statistic analysis section of the Material and Methods), so what has been tested is if bending leads to an increase (or a decrease, but anyway only one direction) in the impedance of the electrode. Please amend. Moreover, shouldn't the test be paired rather than unpaired?

9) Please remove "similar trends are observed also for CD68 and IgG". At 3 and 6 months time there is no clear trend to decrease in CD68 and IgG expression.

Reviewer #2 (Remarks to the Author):

As I summarized in my first review "Flexible array technologies, particularly those with multimodal function, is of keen interest to the research and translational neurotechnology communities. The incorporation of a hydrogel structure builds well on this lab's past work using polymer multimodal fibers, and offers a reduction in shear forces produced by mechanical mismatch between the brain and stiffer electrodes. This body of work shows that hydrogel probes are capable of being inserted into the brain, delivering light to stimulate photosensitive ion channels, delivering pharmacological agents, and recording evoked and spontaneous action potentials. In addition, these experiments demonstrate a reduction in lateral stress produced by simulated brain micromotion. Taken together, this is an exciting body of work demonstrating a technology with the possibility for great benefit for experimental protocols."

The authors have addressed my comments with additional data and analysis, and now demonstrate clear superiority of this hydrogel device over the earlier polymer version. The hydrogel implant now can achieve longer single neuron recording and experiences earlier wound healing as compared to the polymer device, two key metrics of in vivo viability. This is an exciting result, and one that I look forward to seeing in publication!!

I do have some minor comments to the authors, summarized below.

Comment 1 – The data presented comparing the polymer and hydrogel version of the probe is important and compelling. I would encourage the reviewers to include this comparison in the main Figure 3, rather than in Supp Fig 19. In my opinion, this is the single most important figure of the in vivo characterization.

However, despite where the data is presented, I do think that the polymer and hydrogen Vpp amplitude and noise should be presented on the same axis. Having different axis for the two electrode types is confusing at best. Please revise panels a and b to compare the two device types on the same axis. The finding that the voltage amplitude is lower for hydrogel than for polymer is interesting, and likely due to neurons being pushed further away from the device. Notably, the amplitude normalizes to the polymer devices by roughly 8 weeks. This highlights that hydrogel devices are likely preferable for long-term recordings, but may be equivalent to polymer for <4 week duration recording experiments. Using a single y axis will also demonstrate that the noise level of the hydrogel devices is actually much lower, which accounts for the improved SNR starting at ~ 4 weeks compared to the polymer devices.

Comment 2 – The new data presenting additional single units in Sup. Fig 21 is compelling. However, it is unclear how many animals have been recorded from, and how many units are obtained per device. This type of information should be readily available, and would add important context to understanding the performance of these devices. This should be included both in the results and methods sections.

Comment 3 – Similar response as above. The data look good, but it is unclear how many animals/devices/electrodes are included in this dataset – please add these. In addition, for Sup Fig 22. Panel b. please use a single y axis for appropriate comparison.

Comment 4 – Great response. It may be worth drawing the parallel between the increasing amplitude and SNR on the same timescale as the mitigated inflammatory response (ie. Starts to differentiate from polymer probes around 4 weeks). Perhaps these two are related?

Comment 5 – Great.

Comment 6 – Thank you for this additional detail.

Comment 7 – Really excellent and informative.

Comment 8 – please include n=8 animals, and number of channels and units in the results and methods.

Comment 9 – Ok.

Comment 10 – Great.

Comment 11 – Lovely figure and very informative.

Cristin Welle

Reviewer #4 (Remarks to the Author):

This research group is famous for the multifunctional fiber brain probe that can simultaneously perform optical, electrical, and chemical stimulations/measurements, and has published many excellent papers with the similar contents as the present report. The main advance in the present study is the coating of the fibers with a hydrogel to reduce brain damage during insertion and implantation of the probe. The soft nature of hydrogel enabled the long-term use of the probe more than 6 months for freely moving mice.

The idea of hydrogel-coating of the probe has been studied long, and the hydrogel material used here is not new. Therefore, I agree with the comment common to all three reviewers that is "the effect of hydrogel-coating should be clearly shown by direct comparison with the non-coated probe". The Supplementary Fig19 and 22 nicely show the advantage of hydrogel-coating for longer-time stable use of the probe. These supplementary data should be the most important results to be shown in the Figures of the main text. However, it seems a little strange that all of n=8 experiments with uncoated probes lost the signal at 12 weeks. Also, new finding in brain science by the longer-term measurement (6 month) should be discussed.

It is also agreeable that Reviewer 3 request optimization of the material kind and amount of the hydrogel by taking account the biochemical advantage of the hydrogel coating.

Unfortunately, the response from authors are almost limited to the mechanical properties of hydrogel (stiffness and adhesiveness). At least, the coating amount of hydrogel (thickness of the hybrid probe) should be optimized based on experiments.

Response to Reviewer #1:

General comment. *In this revised version, Park et al. improved the text and the figures. The authors addressed some of the points raised in the previous review. The technology described in the paper shows some more convincing potentials for long-term unit recording for neuroscience applications compared to other (polymer) probes (Supplementary Fig. 19, 22). Hydrogel probes also lead to a lower amount of astrocyte and microglia activation (Fig. 4 and supplementary Fig. 23) compared to stainless steel probes. There are, however, some important points that remain to be addressed in the text.*

Response. Thank you for your insightful and constructive comments. To address yours and other reviewers' comments and concerns in full, we have added or revised figures and tables, introduced clarifications, and expanded discussions within our manuscript. In the following paragraphs, we address each comment point by point. Sentences newly added to or revised in the manuscript and the supplementary information are marked in blue.

Comment 1. *It is still unclear to this referee how many channels per probe remain functional over implantation time and how many units can be simultaneously recorded with one probe (see also comment #5 of previous Referee#2). This needs to be explicitly stated in the results and commented in the discussion. It seems that the maximal number of 2 units was recorded from one probe per animal out of 21 recording sites.*

Response 1. We appreciate the reviewer's recommendation to explicitly state about the specific information of recorded units. In our hydrogel hybrid probes, each microelectrode could measure 1 or 2 single units while the electrodes in each microelectrode array fiber (7 electrodes per each fiber, 3 array fibers per probe) could only measure the same pattern due to the nature of connectorization (all electrodes in each fiber shared the same electrical connection, i.e. were shorted together on purpose). Therefore, the total number of single units recorded per probe varied between 2 and 6 in the current study. **The goal of our study was not to maximize the number of units in this particular design, a task much more readily accomplished by the sophisticated and costly CMOS platforms such as Neuropixels, but rather provide a framework for combining multiple functional features while maintaining high-quality recordings and low foreign-body response.** Future collaborative studies will undoubtedly improve upon the channel count by outfitting our platforms with integrated backends, which are not the topic of research in our team. Nevertheless, we appreciate and agree with the reviewer's suggestion to explicitly state the maximum recordable single units of our probe in the manuscript. To clarify these points, we have added the following sentences (Result & Discussion) and the new Supplementary Table 3 in the revised manuscript:

On page 10, the sentence "Each microelectrode fiber in the hydrogel hybrid probe measured 1 or 2 single unit-potentials (spikes). It should be noted that individual electrodes in each microelectrode fiber were not electrically distinguishable due to collective electrical connectorization procedure, and thus they measured the same pattern. This cross-talk was not a feature of the electrode array design or polymer insulation, which were previously shown sufficient

to avoid cross-talk between neighboring channels, but instead was a fabrication choice for this proof-of-concept study.”

On page 11, the sentence “The number of functional channels and the maximum number of recordable single units per probe considering overlapped patterns (between 2 to 6) are provided in the **Supplementary Table 3.**”

On page 12, the sentence “Although the hydrogel hybrid probes offer functional advantages in tissue response, there are several points of further improvement left for the future development and applications. First, the complexity, size, and recording capacity of the microelectrode fibers are limited by the back-end electrical connectorization and resultant overlap in signal patterns from adjacent electrodes in each microelectrode fiber. An improved approach to establish high-resolution connections between functional fibers and back-end interfaces would enhance the throughput of recorded units.”

Supplementary Table 3 | The maximum number of spikes, the number of patterns, and the number of functional electrodes per probe

Max unit numbers (by pattern)	Functional channels considering overlapped pattern
3 (2 / 1 / 0)	2 (14)
4 (2 / 1 / 1)	3 (21)
2 (1 / 1 / 0)	2 (14)
3 (0 / 1 / 2)	2 (14)
6 (2 / 2 / 2)	3 (21)
1 (0 / 1 / 0)	1 (7)
3 (1 / 1 / 0)	3 (21)
3 (0 / 1 / 1)	2 (14)

Comment 2. *Supplementary figure 16: the electrical signals triggered by optogenetic stimulation fall within the illumination window in panel b, but not in panel a. There seems to be a mistake there.*

Response 2. Thank you for pointing out the issues in **Supplementary Fig. 16**. Because of the endogenous characteristics of optogenetically-evoked potential, there is the time-delay (~10 ms) between the onset of optical stimulation and the electrical potential changes. After a thorough inspection of the data, we confirmed that the panel a is correctly presented while the blue bars representing the optical stimulation in panel b appeared (erroneously) longer due to illustration. We have corrected this error in **Supplementary Fig. 16** in the revised manuscript.

Supplementary Fig. 16 | Optically-evoked potentials in a large time scale. a,b, Representative electrophysiological recordings in the vHPC during optical stimulation (10 Hz, 10 mW mm⁻², 5 ms pulse width) from the hydrogel hybrid probes 4 weeks following the implantation and transfection with AAV5-CaMKII α ::ChR2-eYFP with time window of 0.1 second (a) and 1.5 second (b). The orange box in **b** represents the data shown in **a**.

Comment 3. *Supplementary figure 20: the data in panel a, fourth bar from the left (Saline, light ON, eYFP), appear to be the same data shown in panel b, fourth bar from the left (NBQX+AP5, light ON, eYFP). Yet, the two experimental conditions are different. Please comment/amend.*

Response 3. Thank you for pointing out this mistake in **Supplementary Fig. 20 (now Supplementary Fig. 19)**. We believe that the dataset was incorrectly copied from the Excel spreadsheet into Igor for plotting (that is the fourth column from the “drug” dataset was accidentally copied into the “saline” plot). We added the corrected version of the graph in the revised manuscript (in panel a, fourth bar from the left (Saline, light ON, eYFP)).

**Supplementary Fig. 19 | Investigation of locomotor effects of optogenetic stimulation of the BLA-to-vHPC projection circuit. a**,**b**, Mouse velocity in the open field during the optical stimulation (20 Hz, 10 mW $mm^{-2}$, 5 ms pulse width, OFF/ON/OFF cycle, 3 min epoch). Velocities are measured for mice expressing ChR2-eYFP and eYFP control (a) in the absence or (b) following the injection of the combination of glutamate receptor antagonist cocktail (AMPA and NMDA receptor antagonist, NBQX+AP5). No statistical difference is found for both groups during the optical stimulation (two-way ANOVA with Bonferroni multiple comparison test, saline: $p$ = 0.1804, drug: $p \geq 0.9999$). Values in **a**,**b**, represent the mean and the standard deviation ($n$ = 8).

The velocity table used for this calculation is provided below for the reviewers' examination:

| | | | | | | | | | | |
|---|---|---|---|---|---|---|---|---|---|---|
| | ChR2 saline | | | | | | | | | |
| Mouse # | 1 | 2 | 3 | 4 | 5 | 6 | 7 | 8 | average | stdev |
| OFF | 8.20253 | 5.99088 | 3.1499 | 4.92699 | 5.98203 | 6.7568 | 4.5567 | 6.33296 | 5.737349 | 1.527874 |
| ON | 8.60753 | 5.73863 | 4.1432 | 2.58576 | 4.07224 | 5.77328 | 4.12579 | 3.10384 | 4.768784 | 1.90912 |
| OFF | 10.0679 | 4.94138 | 3.57553 | 4.40545 | 4.03882 | 4.79089 | 4.33028 | 4.45003 | 5.075035 | 2.061317 |
| | | | | | | | | | | |
| | ChR2 drug | | | | | | | | | |
| Mouse # | 1 | 2 | 3 | 4 | 5 | 6 | 7 | 8 | average | stdev |
| OFF | 9.70655 | 4.3553 | 6.38873 | 2.20263 | 5.0447 | 3.65175 | 5.63171 | 1.96314 | 4.868064 | 2.496374 |
| ON | 9.70138 | 8.35601 | 6.57778 | 7.6266 | 4.11428 | 3.36408 | 2.75768 | 4.5751 | 5.884114 | 2.540253 |
| OFF | 6.6628 | 6.94454 | 7.3219 | 4.25498 | 4.66281 | 3.47079 | 3.87165 | 3.48726 | 5.084591 | 1.62344 |
| | | | | | | | | | | |
| | EYFP saline | | | | | | | | | |
| Mouse # | 1 | 2 | 3 | 4 | 5 | 6 | 7 | 8 | average | stdev |
| OFF | 9.62358 | 9.48658 | 9.26002 | 9.23287 | 4.5418 | 4.30462 | 5.24419 | 5.39393 | 7.135949 | 2.448963 |
| ON | 11.1378 | 8.25176 | 8.55431 | 7.98869 | 3.92301 | 4.36075 | 4.06589 | 4.4081 | 6.586289 | 2.737905 |
| OFF | 8.33637 | 7.15676 | 9.87022 | 8.29494 | 3.31426 | 4.61901 | 4.9215 | 4.87833 | 6.423924 | 2.3036 |
| | | | | | | | | | | |

	EYFP drug									
Mouse #	1	2	3	4	5	6	7	8	average	stdev
OFF	7.88488	9.60078	8.30327	7.97645	3.46135	5.12422	4.3236	6.04521	6.58997	2.170233
ON	8.78509	12.2764	10.7807	9.51703	3.09013	4.50485	3.31936	4.49093	7.095561	3.64375
OFF	8.53364	8.78209	8.28274	9.27571	2.54419	4.29978	4.11861	4.39208	6.278605	2.684292

Comment 4. *In the statistics section, it is stated that error bars in all graphs represent s.e.m., but in figure legends (e.g., Fig. 1 h-j) it is reported that error bars represent SD.*

Response 4. Thank you for pointing the inconsistency in our statistics section. We have revised the Statistical analysis description in Methods section as the follows and updated the size of error bars to the standard deviation (SD) throughout the revised manuscript:

On page 33, the sentence “Normality of data distribution was tested via Shapiro-Wilk test. For the analysis of the behavioral data that showed normal distribution, two-way ANOVA followed by Bonferroni’s multiple comparison test and Tukey’s multiple comparison test were conducted with threshold of $*p < 0.05$, $**p < 0.01$, $***p < 0.001$. For immunohistochemistry experiments, where data distributions were found to be non-normal, Kruskal-Wallis test followed by Bonferroni’s correction and Dunn-Sidak multiple comparison test were conducted with threshold of $*p < 0.05$. For the fiber characterization, paired two-sided Student’s t-tests were used, and significance threshold was placed at $*p < 0.05$, $**p < 0.01$. Error bars represent standard deviation.”

Comment 5. *Lines 269-270: the decrease in SNR 4 weeks after implantation in polymer probes needs to be statistically quantified.*

Response 5. We appreciate the reviewer’s insightful comment. To clarify this point, we have added the statistical values and edited the following description in the revised manuscript as well as revised the **Supplementary Fig. 21**:

On page 11, the sentence “Furthermore, SNR in the recordings acquired with the hydrogel hybrid probes remains stable at least up to 24 weeks following the implantation, whereas the recordings acquired with polymer probes exhibit substantial decrease in SNR 4 weeks following implantation (**Supplementary Fig. 21**, $p = 0.000578$ and 0.000124 between 4 weeks and 8 or 12 weeks, respectively).”

Supplementary Fig. 21 | Endogenous electrophysiological signals recorded from various probes over time. a-c, Amplitude of endogenous activities (a), background noise (b), and signal-to-noise ratio (SNR) (c) in the opto-electrophysiological experiments recorded from the hydrogel hybrid (orange) and polymer probes (purple) at various time points ($n = 8$ for each, $p = 0.000578$ and 0.000124 between 4 weeks and 8, 12 weeks, respectively).

Comment 6. A normality test must be run before choosing the statistical test to be used (parametric vs non parametric), rather than assuming normality based on previous work.

Response 6. We thank the reviewer for this insight. Following the reviewer's suggestion, we have performed normality tests on our data (Shapiro-Wilk test was employed due to number of samples < 30). As a result, we found that the null hypothesis (normal distribution) is not rejected for an alpha level of 0.05 in the majority of our experiments. However, the immunohistochemistry data showed the non-parametric characteristics, which motivated us to switch from the two-way ANOVA followed by Bonferroni's multiple comparison test with Tukey's multiple comparison test to the Kruskal-Wallis test followed by Bonferroni's correction with Dun-Sidak multiple comparison test in the revised manuscript. This has been reflected in our revised manuscript:

On page 33, the sentence “Normality of data distribution was tested via Shapiro-Wilk test. For the analysis of the behavioral data that showed normal distribution, two-way ANOVA followed by Bonferroni’s multiple comparison test and Tukey’s multiple comparison test were conducted with threshold of $*p < 0.05$, $**p < 0.01$, $***p < 0.001$. For immunohistochemistry experiments, where data distributions were found to be non-normal, Kruskal-Wallis test followed by Bonferroni’s correction and Dunn-Sidak multiple comparison test were conducted with threshold of $*p < 0.05$. For the fiber characterization, paired two-sided Student’s t-tests were used, and significance threshold was placed at $*p < 0.05$, $**p < 0.01$. Error bars represent standard deviation.”

Fig. 4 | Long-term investigation of single-unit potentials and foreign body responses. **a**, Representative spontaneous electrophysiological activity recorded in the vHPC using a hydrogel hybrid probe 1 month following implantation. **b**, Principle component analysis of the single-unit potentials (spikes) recorded 3 days, 1 week, 2 weeks, 1–6 months following implantation. Closed circles indicate centers and ellipses represent two standard deviations (2σ) contours of the principle component distributions. **c**, Average spike waveforms recorded over time corresponding to clusters in (b). **d**, Average amplitude (red) and noise level (blue) of spikes in (c) recorded over time. **e**, Average inter-spike intervals for each unit. Significance is confirmed by one-way Student's t test

(*** $p < 0.001$). **f-i**, Average immunofluorescence area quantifying the presence of GFAP (f), Iba1 (g), CD68 (h), and IgG (i) in the vicinity of the hydrogel hybrid probes, polymer probes, and stainless steel microwires for 3 days, 1 week, 1 month, 3 months, and 6 months following implantation, respectively. Values in **f-i** represent the mean and the standard deviation ($n = 6$ for each device and time point, Kruskal-Wallis test and Dun-Sidak multiple comparison test ($*p < 0.05$)).

Comment 7. Line 265 and lines 234-235: please specify the statistical test used.

Response 7. Thank you for your comment regarding the statistical test. We have specified the statistical test we used for each experiment in the revised manuscript:

On page 9, the sentence “Consistent with previous studies of the BLA-to-vHPC projection circuit, optical stimulation (20 Hz, 5 ms pulse width, 10 mW mm⁻²) of the BLA inputs in the vHPC delivered via the hybrid probes results in reduction of the time spent in the center of the open field by mice expressing ChR2-eYFP as compared to eYFP controls (two-way ANOVA and Bonferroni multiple comparison test, $p = 0.0031$) (Fig. 3h,j,k).”

On page 10, the sentence “The different units further maintain their characteristics and statistically distinct inter-spike intervals (one-way Student’s t-test, $p < 0.001$) (Fig. 4e).”

Comment 8. Fig. 11: the text reports that the electrical properties of the hybrid device are not significantly affected (either increased or decreased, both directions) by bending. However, the test is one sided (as stated in the statistic analysis section of the Material and Methods), so what has been tested is if bending leads to an increase (or a decrease, but anyway only one direction) in the impedance of the electrode. Please amend. Moreover, shouldn’t the test be paired rather than unpaired?

Response 8. Thank you for your insightful comment. We have used paired two-sided Student’s t test for statistical analysis of fiber characterization in the revised manuscript:

On page 17, caption Fig. 1i “Tip impedance of the electrodes within the fiber arrays in the hydrogel hybrid probes at 0° and 90° bending (paired two-sided Student’s t-test: $p = 0.5232$).”

On page 33, the sentence “For the fiber characterization, paired two-sided Student’s t-tests were used, and significance threshold was placed at $*p < 0.05$, $**p < 0.01$. Error bars represent standard deviation.”

Comment 9. Please remove “similar trends are observed also for CD68 and IgG”. At 3 and 6 months time there is no clear trend to decrease in CD68 and IgG expression.

Response 9. We appreciate the reviewer’s suggestion about the clearer explanation for our immune-histochemistry experiments. In the revised manuscript, we removed the following

sentence: “~~While the differences in GFAP and Iba1 expression are statistically significant, similar trends are also observed for CD68 and IgG.~~”

We greatly appreciate the reviewer’s time and invaluable suggestions which have significantly improved the work. We are particularly grateful to the reviewer for identifying errors in our data presentation as well as suggestions for improving our statistical analyses. We have learned a lot and we sincerely hope that our revised manuscript fully addresses the reviewers’ concerns.

Response to Reviewer #2:

General comment. *As I summarized in my first review "Flexible array technologies, particularly those with multimodal function, is of keen interest to the research and translational neurotechnology communities. The incorporation of a hydrogel structure builds well on this lab's past work using polymer multimodal fibers, and offers a reduction in shear forces produced by mechanical mismatch between the brain and stiffer electrodes. This body of work shows that hydrogel probes are capable of being inserted into the brain, delivering light to stimulate photosensitive ion channels, delivering pharmacological agents, and recording evoked and spontaneous action potentials. In addition, these experiments demonstrate a reduction in lateral stress produced by simulated brain micromotion. Taken together, this is an exciting body of work demonstrating a technology with the possibility for great benefit for experimental protocols."*

The authors have addressed my comments with additional data and analysis, and now demonstrate clear superiority of this hydrogel device over the earlier polymer version. The hydrogel implant now can achieve longer single neuron recording and experiences earlier wound healing as compared to the polymer device, two key metrics of in vivo viability. This is an exciting result, and one that I look forward to seeing in publication!!

I do have some minor comments to the authors, summarized below.

Response. We are delighted that the reviewer has found our work important and exciting. To make our manuscript clearer for the readers, in the following paragraphs, we address each comment point-by-point. Sentences newly added to or revised in the manuscript and the supplementary information are marked in blue.

Comment 1. *The data presented comparing the polymer and hydrogel version of the probe is important and compelling. I would encourage the reviewers to include this comparison in the main Figure 3, rather than in Supp Fig 19. In my opinion, this is the single most important figure of the in vivo characterization.*

However, despite where the data is presented, I do think that the polymer and hydrogen Vpp amplitude and noise should be presented on the same axis. Having different axis for the two electrode types is confusing at best. Please revise panels a and b to compare the two device types on the same axis. The finding that the voltage amplitude is lower for hydrogel than for polymer is interesting, and likely due to neurons being pushed further away from the device. Notably, the amplitude normalizes to the polymer devices by roughly 8 weeks. This highlights that hydrogel devices are likely preferable for long-term recordings, but may be equivalent to polymer for <4 week duration recording experiments. Using a single y axis will also demonstrate that the noise level of the hydrogel devices is actually much lower, which accounts for the improved SNR starting at ~ 4 weeks compared to the polymer devices.

Response 1. We appreciate the reviewer's recommendations how to improve our data presentation. Following these suggestions, now we moved the previous **Supplementary Fig. 19** to Main Manuscript **Fig. 3**. Furthermore, the data from both the hydrogel and polymer probes are

now presented on the same axis. Also, the following sentences have also been revised in our manuscript:

On page 9, the sentence “Potentials correlated with laser pulses are observed 1-2 weeks after the virus injection, and their amplitudes gradually increase up to 8 weeks, at which point they reach a stable level maintained for at least 4 more months (Fig. 3d). No light-evoked potentials are recorded in mice injected with the control virus (Supplementary Fig. 18). Consistent with prior studies³⁵, the noise level decreases 2 weeks following the implantation and is maintained for at least 6 months (Fig. 3e). Signal-to-noise ratio (SNR) gradually increases over the first 4 weeks following the implantation and then plateaus for another 5 months, in contrast to the gradually decreasing SNR recorded from the polymer probes⁵ (Fig. 3f).”

Fig. 3 | Optogenetic interrogation by hydrogel hybrid probes. **a**, A photograph of a mouse implanted with the hydrogel hybrid probe. Scale bar: 2 cm. **b**, An illustration of optogenetic modulation and electrophysiological recording in the BLA-to-vHPC projection circuit. **c**, Representative electrophysiological recordings in the vHPC during optical stimulation (10 Hz, 10 mW mm⁻², 5 ms pulse width) using the hydrogel hybrid probes 3 days, 1–2 weeks, and 1–6 months following implantation and transfection with AAV5-CaMKII α ::ChR2-eYFP. **d-f**, Amplitude of optically-evoked potentials (d) and background noise (e), and signal-to-noise ratio (SNR) (f) in the opto-electrophysiological experiments recorded from the hydrogel hybrid (red) and polymer probes (blue) at various time points ($n = 8$ for each). **g**, A summary of the behavioral assays performed in this study. Open field tests (OFT) are performed (3 min light OFF/ON/OFF epoch) 6 weeks following the injection of AAV5-CaMKII α ::ChR2-eYFP or AAV5-CaMKII α ::eYFP. Time spent in the center region of the OFT arena during optical stimulation (20 Hz, 10 mW mm⁻², 5 ms pulse width) is analyzed for trials conducted with the injections of saline solution or glutamate receptor antagonist cocktail (NBQX+AP5). Optical stimuli and drug injections are delivered into the vHPC via the hydrogel hybrid probes. **h**, Time spent in the center in the absence or presence of optical stimulation following injection of saline solution (two-way ANOVA and Bonferroni multiple comparison test, $p = 0.0031$) **i**, Time spent in the center in the absence or presence of optical stimulation following injection glutamate receptor antagonist cocktail ($p > 0.9999$). **j-m**, Representative heat map images tracing the position of mice transfected with ChR2 (j,l) and eYFP (k,m) control with injection of saline solution (j,k) and glutamate receptor antagonist cocktail (l,m). Dashed line indicates the center region of the open field arena. Values in **d-f,h,i** represent the mean and the standard deviation ($n = 8$).

Comment 2. *The new data presenting additional single units in Sup. Fig 21 is compelling. However, it is unclear how many animals have been recorded from, and how many units are obtained per device. This type of information should be readily available, and would add important context to understanding the performance of these devices. This should be included both in the results and methods sections.*

Response 2. Thanks for your insightful comment from the reviewer about the number of animals and recorded units per probe. We used 8 animals (1 probe per animal), and the presenting data is average value of one electrode from different probes. Multiple electrodes in a single device are not considered to exclude the device-dependency. We agree that specific information about recording performance of our devices is critical to their future applications. Therefore, we have added the Supplementary Table 3 and revised our manuscript accordingly.

Briefly, in our hydrogel hybrid probes, each microelectrode could measure 1 or 2 single units while the electrodes in each microelectrode array fiber (7 electrodes per each fiber, 3 array fibers per probe) could only measure the same pattern due to the nature of connectorization (all electrodes in each fiber shared the same electrical connection, i.e. were shorted together on purpose). Therefore, the total number of single units recorded per probe varied between 2 and 6 in the current study. **The goal of our study was not to maximize the number of units in this particular design, a task much more readily accomplished by the sophisticated and costly CMOS platforms such as Neuropixels, but rather provide a framework for combining multiple functional features while maintaining high-quality recordings and low foreign-body response.** Future

collaborative studies will undoubtedly improve upon the channel count by outfitting our platforms with integrated backends, which are not the topic of research in our team.

On page 10, the sentence “Each microelectrode fiber in the hydrogel hybrid probe measured 1 or 2 single unit-potentials (spikes). It should be noted that individual electrodes in each microelectrode fiber were not electrically distinguishable due to collective electrical connectorization procedure, and thus they measured the same pattern. This cross-talk was not a feature of the electrode array design or polymer insulation, which were previously shown sufficient to avoid cross-talk between neighboring channels, but instead was a fabrication choice for this proof-of-concept study.”

On page 11, the sentence “The number of functional channels and the maximum number of recordable single units per probe considering overlapped patterns (between 2 to 6) are provided in the **Supplementary Table 3**.”

On page 12, the sentence “Although the hydrogel hybrid probes offer functional advantages in tissue response, there are several points of further improvement left for the future development and applications. First, the complexity, size, and recording capacity of the microelectrode fibers are limited by the back-end electrical connectorization and resultant overlap in signal patterns from adjacent electrodes in each microelectrode fiber. An improved approach to establish high-resolution connections between functional fibers and back-end interfaces would enhance the throughput of recorded units.”

On page 31, the sentence “In the signal analysis of the device electrical performance (amplitude, noise, and SNR), 8 animals were used (1 probe implanted per animal) and the average value of one electrode from different probe was used. Multiple electrodes in a single device were not considered to exclude the device-dependency.”

Supplementary Table 3 | The maximum number of spikes, the number of patterns, and the number of functional electrodes per probe

Max unit numbers (by pattern)	Functional channels considering overlapped pattern
3 (2 / 1 / 0)	2 (14)
4 (2 / 1 / 1)	3 (21)
2 (1 / 1 / 0)	2 (14)
3 (0 / 1 / 2)	2 (14)
6 (2 / 2 / 2)	3 (21)
1 (0 / 1 / 0)	1 (7)
3 (1 / 1 / 0)	3 (21)
3 (0 / 1 / 1)	2 (14)

Comment 3. *Similar response as above. The data look good, but it is unclear how many animals/devices/electrodes are included in this dataset – please add these. In addition, for Sup Fig 22. Panel b. please use a single y axis for appropriate comparison.*

Response 3. We appreciate this insightful suggestion. Now all the data in the revised manuscript include the information about the number of animals, devices, and electrodes, and more specific information is added with **Supplementary Table 3** as discussed in the **Response 2**. We also amended the Supplementary Figure 21 to the one with single y axis for appropriate comparison:

Supplementary Fig. 21 | Endogenous electrophysiological signals recorded from various probes over time. a-c, Amplitude of endogenous activities (a), background noise (b), and signal-to-noise ratio (SNR) (c) in the opto-electrophysiological experiments recorded from the hydrogel hybrid (orange) and polymer probes (purple) at various time points ($n = 8$ for each, $p = 0.000578$ and 0.000124 between 4 weeks and 8, 12 weeks, respectively).

Comment 4. *Great response. It may be worth drawing the parallel between the increasing amplitude and SNR on the same timescale as the mitigated inflammatory response (ie. Starts to differentiate from polymer probes around 4 weeks). Perhaps these two are related?*

Response 4. We appreciate the reviewer’s suggestion to investigate the relationship between the inflammatory response and the quality of the recorded signals. Following the suggestion, we plotted the GFAP expression (representing astrocytic accumulation) on the same graphs as the amplitude or SNR of the recorded signals. There appears to be a relationship between the reduced astrocytic scarring and improved SNR in the recordings acquired with hydrogel hybrid probes. While these data may not fully capture the dynamic phenomena, our data provides additional

evidence that the degree of scarring negatively affects recording utility. Hence, we have added this analysis as **Supplementary Fig. 24** in the revised manuscript:

On page 11, the sentence “At longer post-implantation time points (1 month, 3 months, and 6 months), the hydrogel hybrid probes elicit substantially reduced foreign body response as compared to both polymer probes and steel microwires (**Fig. 4f-i**), which is correlated with the higher SNR of the recorded signals (**Supplementary Fig. 24**).”

Supplementary Fig. 24 | Relationship between foreign body response and quality of the recorded-signal. a,b, Comparison between the average immunofluorescence area quantifying the presence of GFAP in the vicinity of the hydrogel hybrid (gray) and polymer probes (pink) and the amplitude (a) and the signal-to-noise ratio (SNR) (b) of the optically-evoked activities recorded with hydrogel hybrid (black) and the polymer probes (red) at various time points (1 week and 1, 3, 6 months, $n = 6$ for immuno-histochemical experiment and $n = 8$ for opto-electrophysiological experiment).

Comment 5. Great.

Comment 6. Thank you for this additional detail.

Comment 7. Really excellent and informative.

Response 5-7. We greatly appreciate the reviewer’s positive comments.

Comment 8. please include $n=8$ animals, and number of channels and units in the results and methods.

Response 8 Thanks for your insightful suggestion. Now all the data in the revised manuscript include the information about the number of animals, devices, and electrodes, and more specific information is added with **Supplementary Table 3** as discussed in the **Response 2**.

Comment 9. Ok.

Comment 10. Great.

Comment 11. *Lovely figure and very informative.*

Response 9-11. We greatly appreciate the reviewer's enthusiasm about our revision letter.

We greatly appreciate the reviewer's insight and invaluable suggestions which have significantly strengthened our manuscript. We sincerely hope the revised manuscript fully addresses the reviewer's concerns.

Response to Reviewer #3:

Comment 1. *This research group is famous for the multifunctional fiber brain probe that can simultaneously perform optical, electrical, and chemical stimulations/measurements, and has published many excellent papers with the similar contents as the present report. The main advance in the present study is the coating of the fibers with a hydrogel to reduce brain damage during insertion and implantation of the probe. The soft nature of hydrogel enabled the long-term use of the probe more than 6 months for freely moving mice.*

The idea of hydrogel-coating of the probe has been studied long, and the hydrogel material used here is not new. Therefore, I agree with the comment common to all three reviewers that is "the effect of hydrogel-coating should be clearly shown by direct comparison with the non-coated probe". The Supplementary Fig19 and 22 nicely show the advantage of hydrogel-coating for longer-time stable use of the probe. These supplementary data should be the most important results to be shown in the Figures of the main text. However, it seems a little strange that all of n=8 experiments with uncoated probes lost the signal at 12 weeks. Also, new finding in brain science by the longer-term measurement (6 month) should be discussed.

Response 1. We appreciate the reviewer's recognition of importance of our work and their thoughtful evaluation of our manuscript. Following the reviewer's suggestion, we now moved the previous **Supplementary Fig. 19** to Main Manuscript **Fig. 3**. Moreover, the single y-axis is used for the graphs in **3d-f** to appropriately compare the signals acquired with the polymer and the hydrogel hybrid probes. While the polymer probes remained within animal brains 12 weeks after the implantation, we stopped the recording from these probes as the low signal-to-noise ratio no longer permitted identification of neural data ($\text{SNR} \leq 1$).

On page 9, the sentence "Potentials correlated with laser pulses are observed 1-2 weeks after the virus injection, and their amplitudes gradually increase up to 8 weeks, at which point they reach a stable level maintained for at least 4 more months (**Fig. 3d**). No light-evoked potentials are recorded in mice injected with the control virus (**Supplementary Fig. 18**). Consistent with prior studies³⁵, the noise level sharply decreases 2 weeks following the implantation and is maintained for at least 6 months (**Fig. 3e**). Signal-to-noise ratio (SNR) gradually increases over the first 4 weeks following the implantation and then plateaus for another 5 months, in contrast to the gradually decreasing SNR recorded from the polymer probes⁵ (**Fig. 3f**)." is highlighted in blue.

Fig. 3 | Optogenetic interrogation by hydrogel hybrid probes. **a**, A photograph of a mouse implanted with the hydrogel hybrid probe. Scale bar: 2 cm. **b**, An illustration of optogenetic modulation and electrophysiological recording in the BLA-to-vHPC projection circuit. **c**, Representative electrophysiological recordings in the vHPC during optical stimulation (10 Hz, 10 mW mm⁻², 5 ms pulse width) using the hydrogel hybrid probes 3 days, 1–2 weeks, and 1–6 months following implantation and transfection with AAV5-CaMKII α ::ChR2-eYFP. **d–f**, Amplitude of optically-evoked potentials (**d**) and background noise (**e**), and signal-to-noise ratio (SNR) (**f**) in the opto-electrophysiological experiments recorded from the hydrogel hybrid (red) and polymer probes (blue) at various time points ($n = 8$ for each). **g**, A summary of the behavioral assays performed in this study. Open field tests (OFT) are performed (3 min light OFF/ON/OFF epoch) 6 weeks following the injection of AAV5-CaMKII α ::ChR2-eYFP or AAV5-CaMKII α ::eYFP. Time spent in the center region of the OFT arena during optical stimulation (20 Hz, 10 mW mm⁻²)

², 5 ms pulse width) is analyzed for trials conducted with the injections of saline solution or glutamate receptor antagonist cocktail (NBQX+AP5). Optical stimuli and drug injections are delivered into the vHPC via the hydrogel hybrid probes. **h**, Time spent in the center in the absence or presence of optical stimulation following injection of saline solution (two-way ANOVA and Bonferroni multiple comparison test, $p = 0.0031$) **i**, Time spent in the center in the absence or presence of optical stimulation following injection glutamate receptor antagonist cocktail ($p > 0.9999$). **j-m**, Representative heat map images tracing the position of mice transfected with ChR2 (j,l) and eYFP (k,m) control with injection of saline solution (j,k) and glutamate receptor antagonist cocktail (l,m). Dashed line indicates the center region of the open field arena. Values in **d-f,h,i** represent the mean and the standard deviation ($n = 8$).

Furthermore, we appreciate the reviewer's comment on the fundamental studies that our technology may enable. To reflect this, we have added the following sentence in the Discussion of the revised manuscript:

On page 13, the sentence “Taken together, the advances demonstrated in this work may open opportunities for future studies of neurobiological phenomena requiring wider experimental time windows (> 6 months) including development and aging, and fundamental understanding of progressive neurological disorders such as Parkinson's disease and Alzheimer's disease.”

Comment 2. *It is also agreeable that Reviewer 3 request optimization of the material kind and amount of the hydrogel by taking account the biochemical advantage of the hydrogel coating. Unfortunately, the response from authors are almost limited to the mechanical properties of hydrogel (stiffness and adhesiveness). At least, the coating amount of hydrogel (thickness of the hybrid probe) should be optimized based on experiments.*

Response 2. Thank you for your insightful comment. In this work, we have focused on mechanical properties of hydrogels and their effect on longevity of the implanted probes as our primary design parameter for optimization was mechanical compliance of the hydrogel hybrid probes (bending stiffness). We appreciate the reviewer's comment that biochemical advantages of hydrogel matrix can provide further benefits to our proposed design. To discuss this potential benefit for the future study, we have added the following sentence in Discussion of the revised manuscript:

On page 12, the sentence “Also, while here we focus on mechanical aspects of hydrogel-hybrid probe biocompatibility and longevity, this platform may offer additional strategies for improving long-term performance. For instance, incorporation of biofunctional substances such as growth factors, neuroattractants, and cell-adhesion enhancers within the hydrogel matrix may further offer biochemical advantages to promote the intimate integration of the tissue with the probe elements³⁶⁻³⁸.”

Furthermore, we have developed a fabrication method for the proposed hydrogel hybrid probe (**Supplementary Fig. 2**) based on a micro laser-machined polyimide scraper (**Supplementary Fig. 3**) to ensure the uniform thickness of hydrogel matrix around the fiber assembly (**Supplementary Fig. 11**). To clearly indicate this fabrication method and provide the thickness of the hydrogel matrix, we have added the following information in the revised manuscript:

On page 5, the sentence “To avoid bulk hydrogel layer formation around the hybrid probes, any excessive pre-gel solution on the probe is scrapped off by a polyimide scraper prior to polymerization (**Supplementary Fig. 3b**) to ensure the uniform thickness (25 μm) of hydrogel matrix around the polymer fiber assembly.”

Supplementary Fig. 2 | Illustration for fabrication of hydrogel hybrid probes. (1) Individual functional fibers fabricated via thermal drawing are assembled within a polyimide guide fixture, and then connected to the external terminals (optical ferrules, electrical pin connectors, and fluidic tubing). The probes used in this study consist of one waveguide at the center (PC core and COC cladding, $105.9 \pm 8.0 \mu\text{m}$ diameter), three microelectrode arrays (7 Tin microwire ($4.75 \pm 2.22 \mu\text{m}$ diameter) encapsulated PEI insulating cladding ($80.0 \pm 1.8 \mu\text{m}$ diameter), and three microfluidic channels (PEI microtube, $54.0 \pm 2.1 \mu\text{m}$ inner and $115.4 \pm 3.0 \mu\text{m}$ outer diameters) in an alternating concentric arrangement; (2) The surfaces of the fibers within the assembly are functionalized with primary amines and subsequently alginate, and then dip-coated with the hydrogel pre-gel solution; (3) Excessive pre-gel solution is scrapped off with a polyimide scraper to ensure the uniform thickness (25 μm) of hydrogel matrix around the polymer fiber assembly; (4) The hydrogel pre-gel solution is cured by ultraviolet (UV) irradiation to form a soft tough hydrogel matrix; (5) The

resultant hydrogel hybrid probe is thoroughly washed with PBS before implantation to remove unreacted reagents.

Supplementary Fig. 3 | A guide fixture and a scraper for the fabrication of hydrogel hybrid probes. **a**, Guide fixture for functional fiber assembly. Scale bar: 2 mm. **b**, Scraper for the excess hydrogel removal following the hydrogel dip-coating process to ensure the uniform thickness of hydrogel matrix around the polymer fiber assembly (hydrogel thickness: 25 μm). Scale bar: 5 mm.

Supplementary Fig. 11 | Swelling and dimension of hydrogel hybrid probes. **a**, Swelling of a dried hydrogel hybrid probe within 10 min in DPBS. Scale bars: 200 μm . **b**, Diameter of the dry and swollen hydrogel hybrid probes as a function of distance from the probe tip. Scale bar: 200 μm . Values in **b** represent the mean and the standard deviation ($n = 4$).

We greatly appreciate the reviewer's dedication and insightful suggestions which have significantly improved our manuscript. We sincerely hope the revised manuscript fully addresses the reviewer's concerns.

REVIEWER COMMENTS

Reviewer #1 (Remarks to the Author):

The authors convincingly responded to all my concerns and made appropriate changes in the text and figures. I am supportive of the publication of the manuscript.

Minor

In Supplementary table 3, it is not completely clear what the numbers in parentheses mean in both columns. Please add an additional short explanation in the table legend. Moreover, in the table legend: "The maximum number of spikes" should, I believe, read "The maximum number of units".

Referee's name: Tommaso Fellin

Reviewer #2 (Remarks to the Author):

In this revision, the authors have fully addressed all my concerns. I am very enthusiastic about the revised version and can't wait to see this published.

Reviewer #4 (Remarks to the Author):

Overall, I think the manuscript has been revised well.

As I noted in the previous review, The main advance in the present study is the coating of the fibers with a hydrogel to reduce brain damage during insertion and implantation of the probe. Therefore, it is nice the previous Supplementary Fig. 19 was moved to Main Manuscript Fig. 3 to show the direct comparison with the non-coated probe.

1. It is recommended to explain the reason why the recordings with non-coated probes were stopped 12 weeks after implantation using the criteria ($SNR < 1$) with suitable references (reference 5?).
2. It is also recommended to explain the reason why the thickness of the hydrogel coating was set to 25 μm . Was it optimized thickness?

Response to Reviewer #1:

Comment. *The authors convincingly responded to all my concerns and made appropriate changes in the text and figures. I am supportive of the publication of the manuscript.*

Minor

In Supplementary table 3, it is not completely clear what the numbers in parentheses mean in both columns. Please add an additional short explanation in the table legend. Moreover, in the table legend: “The maximum number of spikes” should, I believe, read “The maximum number of units”.

Response. We appreciate the reviewer’s positive outlook on our manuscript. To address the minor issues brought up by the reviewer, we added an additional brief explanation of the numbers shown in parentheses in table 3 within the corresponding legend as well as changed the word “spikes” to “units” as follows:

Supplementary Table 3 | The maximum number of units and the number of functional electrodes per probe. The digits in the left parentheses represent the number of units recorded for each electrode. The digits in the right parentheses represent the number of working electrodes per probe.

We thank the reviewer for constructive comments throughout the review process.

Response to Reviewer #2:

Comment. *In this revision, the authors have fully addressed all my concerns. I am very enthusiastic about the revised version and can't wait to see this published.*

Response. We appreciate reviewer's enthusiasm about our work and are grateful for the insightful comments throughout the revision process.

Response to Reviewer #4:

General Comment. *Overall, I think the manuscript has been revised well.*

As I noted in the previous review, The main advance in the present study is the coating of the fibers with a hydrogel to reduce brain damage during insertion and implantation of the probe. Therefore, it is nice the previous Supplementary Fig. 19 was moved to Main Manuscript Fig. 3 to show the direct comparison with the non-coated probe.

Response. We appreciate the reviewer's support in our manuscript. To address the minor issues that the reviewer mentioned, we added more specific explanation with proper references in the manuscript.

Comment 1. *It is recommended to explain the reason why the recordings with non-coated probes were stopped 12 weeks after implantation using the criteria (SNR < 1) with suitable references (reference 5?).*

Response 1. We revised the sentences to include the specific results from the uncoated polymer probe as well as the reasoning of stopped signal-recording with suitable citation (reference 5 and 12)

p.9: "Signal-to-noise ratio (SNR) for the hydrogel hybrid probe gradually increases over the first 4 weeks following the implantation and then plateaus for another 5 months (Fig. 3f). In contrast, the SNR recorded from the polymer probes exhibits gradual decrease and becomes one after 12 weeks following implantation due likely due to a greater foreign body caused by a mechanical mismatch with the surrounding neural tissue^{5,12} (**Fig. 3f**)."

Comment 2. *It is also recommended to explain the reason why the thickness of the hydrogel coating was set to 25 μm . Was it optimized thickness?*

Response 2. It has been studied that cells start to sense the mechanical modulus of the underlying substrate when the hydrogel's thickness becomes very thin (below 10-20 μm) (Buxboim, Amnon, et al. "How deeply cells feel: methods for thin gels." *Journal of Physics: Condensed Matter* 22.19 (2010): 194116.). Hence, in this work, we selected 25 μm hydrogel thickness to minimize increase in the size of the hybrid probe while providing sufficient mechanical cloak for the underlying stiffer polymer fibers from the surrounding cells. To clarify this point, we have added the following sentence in the revised manuscript:

p.5: "The hydrogel thickness (25 μm) in the hybrid probes is selected to minimize the size of the devices while providing sufficient mechanical cloaking for the underlying stiffer polymer fibers²⁷."

We thank the reviewer for constructive comments throughout the review process of this work.